# Listwise Preference Diffusion Optimization for User Behavior Trajectories Prediction

**Hongtao Huang**[*]
University of New South Wales
hongtao.huang@unsw.edu.au

**Chengkai Huang**[*]
University of New South Wales
and Macquarie University
chengkai.huang1@unsw.edu.au

**Junda Wu**
University of California San Diego
juw069@ucsd.edu

**Tong Yu**
Adobe Research
tyu@adobe.com

**Julian McAuley**
University of California San Diego
jmcauley@ucsd.edu

**Lina Yao**
CSIRO's Data61
and University of New South Wales
lina.yao@unsw.edu.au

## Abstract

Forecasting multi-step user behavior trajectories requires reasoning over structured preferences across future actions, a challenge overlooked by traditional sequential recommendation. This problem is critical for applications such as personalized commerce and adaptive content delivery, where anticipating a user's complete action sequence enhances both satisfaction and business outcomes. We identify an essential limitation of existing paradigms: their inability to capture global, listwise dependencies among sequence items. To address this, we formulate *User Behavior Trajectory Prediction* (UBTP) as a new task setting that explicitly models long-term user preferences. We introduce *Listwise Preference Diffusion Optimization* (LPDO), a diffusion-based training framework that directly optimizes structured preferences over entire item sequences. LPDO incorporates a Plackett–Luce supervision signal and derives a tight variational lower bound aligned with listwise ranking likelihoods, enabling coherent preference generation across denoising steps and overcoming the independent-token assumption of prior diffusion methods. To rigorously evaluate multi-step prediction quality, we propose the task-specific metric: Sequential Match (SeqMatch), which measures exact trajectory agreement, and adopt Perplexity (PPL), which assesses probabilistic fidelity. Extensive experiments on real-world user behavior benchmarks demonstrate that LPDO consistently outperforms state-of-the-art baselines, establishing a new benchmark for structured preference learning with diffusion models.

## 1 Introduction

Understanding and forecasting user behavior is a vital problem in personalized AI and interactive systems [1, 2, 3, 4, 5, 6]. Many approaches, such as sequential and dynamic recommendation models, use a user's past interaction sequence to predict the next item or action of interest [7, 8].

---

[*]Equal contribution.

39th Conference on Neural Information Processing Systems (NeurIPS 2025).

Traditional sequential recommendation models typically focus on predicting a user's next item based on recent history [9, 10, 11]. Although this one-step (myopic) approach can capture short-term preferences, it often optimizes immediate engagement metrics (e.g. clicks or quick rewards) at the expense of long-term user satisfaction [12, 13]. In fact, immediate feedback signals offer limited insight into a user's lasting interests and can even mislead the system. For example, click-based optimization may favor clickbait content that hurts long-term enjoyment [7]. As a result, next-item recommenders tend to be short-sighted or greedy. Studies have observed that approaches which optimize only short-term rewards (such as contextual bandits for clicks) can yield suboptimal recommendations in the long run [7]. In summary, one-step sequential models often neglect the longer-term consequences of each recommendation, potentially degrading user experience over time.

In contrast to that, predicting user behavior trajectories goes beyond next-item recommendation by forecasting a user's sequence of future actions across multiple time steps or stages. This can involve predicting a user's engagement over a longer horizon (days or weeks ahead), or modeling multi-stage decision processes rather than a single step. By anticipating user behaviors at multiple future timestamps, we can aim to capture longer-term preferences, intentions, and patterns that traditional sequential models might miss. Thus, we introduce User Behavior Trajectory Prediction (UBTP), which challenges a model to generate a coherent, ordered sequence of future interactions based on a user's past behavior. Unlike single-step forecasting, UBTP must account for drifting preferences, capture dependencies among successive actions, and manage the compounding uncertainty inherent in multi-step predictions.

Recently, diffusion models have emerged as a powerful paradigm for generative recommendation [14, 15, 16, 17], providing a probabilistic framework to systematically model uncertainty and produce diversified recommendations through explicit learning of latent data distributions. However, while current diffusion-based recommenders demonstrate effectiveness in next-item prediction scenarios, their direct extension to the user trajectory prediction tasks fails to account for the dynamic evolution of user preferences across different temporal stages within predicted trajectories. This limitation leads to compromised prediction fidelity, where the learned plausible distributions may inadvertently incorporate more non-target items due to unmodeled preference transitions as shown in Figure 1 (b).

Existing diffusion-based recommenders typically incorporate *point-wise* preference signals by adjusting the likelihood of each item independently during denoising, which encourages the model to favor relevant items at individual positions. However, this approach neglects the joint dependencies and relative ordering among items in the final list, crucial aspects for generating coherent multi-item trajectories, and thus can still scatter probability mass over unrelated targets. In contrast, a *list-wise* treatment of preference supervision directly maximizes the joint likelihood of the entire ordered list (cf. Figure 1 (d)), naturally capturing both item relevance and inter-item relationships.

*A natural question arises: can we inject listwise preference supervision directly into the diffusion process to produce more accurate and coherent user behavior trajectories?*

We propose Listwise Preference Diffusion Optimization (LPDO), a novel method that aligns the reverse diffusion process with listwise ranking objectives. The standard training objective of diffusion models optimizes a variational evidence lower bound (ELBO) with marginal reconstruction losses, but it treats each prediction independently and overlooks the ranking structure that is fundamental to personalized user trajectory prediction. Therefore, we inject a Plackett–Luce ranking signal into the variational lower bound, ensuring that at every denoising step the model favors true items over alternatives. This principled integration bridges diffusion denoising and personalized ranking, making the generation process more consistent with the UBTP objective. Compared to non-diffusion models (cf. Figure 1 (a)) and traditional diffusion models (cf. Figure 1 (b)), our preference-aware framework (cf. Figure 1 (c)) produces coherent, preference-aligned trajectory generations. Our main contributions are summarized as follows:

- We formally define User Behavior Trajectory Prediction (UBTP) as the task of forecasting an ordered sequence of k future user interactions, highlighting its practical importance and challenges.

- We propose the Listwise Preference Diffusion Optimization (LPDO), a novel training paradigm that seamlessly incorporates Plackett–Luce listwise ranking into the diffusion ELBO, enabling coherent and preference-aware generation of user behavior trajectories.

- We propose a principled ELBO that incorporates Plackett–Luce ranking terms, tightly coupling diffusion denoising with multi-step ranking likelihoods.

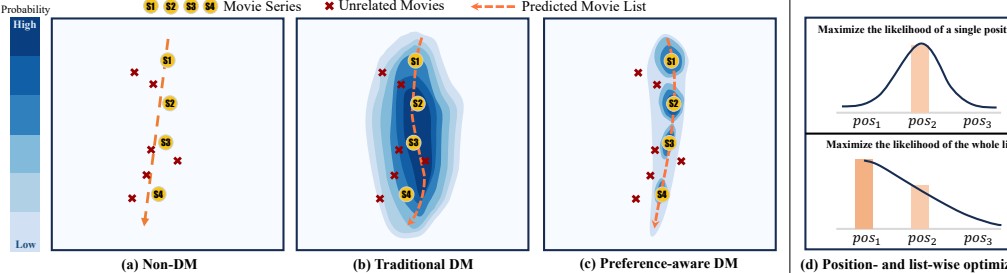

Figure 1: Examples of user trajectory predictions and a comparison of optimization strategies. The circles Ⓢ① Ⓢ② Ⓢ③ Ⓢ④ represent a series of theme-related movies (e.g., Harry Potter film series), and the cross ✖ indicate unrelated movies that user does not prefer; the dash line ←‑‑ denotes the predicted movie list that user might be interested in; the left color bar shows spatial probability. **(a)** Non-DM model (e.g., SASRec [18]) is typically deterministic and predict a fixed trajectory, which often fails to capture users' latent preferences. **(b)** Traditional diffusion model captures a preference distribution to produce a more robust trajectory, but the distribution may overlap with unrelated targets. **(c)** Preference-aware diffusion model incorporates user preference into the sampling process, concentrating the trajectory distribution on related targets and yielding more coherent recommendation lists. **(d)** Comparison of optimization objectives: position-wise preference optimization (top) independently maximizes each position's likelihood and ignores inter-item dependencies; list-wise preference optimization (bottom) maximizes the joint likelihood of the entire ordered list, better capturing ordering and dependencies among items and producing more consistent, accurate behavior trajectories.

- We propose the novel Sequential Match (SeqMatch) metric to rigorously evaluate multi-step prediction quality.

- On four benchmark datasets, LPDO achieves new state-of-the-art performance, substantially improving both precision and sequence coherence over other baselines.

## 2 Related Work.

**User Behavior Modeling.** User behavior modeling plays an important role in recommender systems. Traditional sequential recommendation models [19, 18, 9, 10] focus on predicting the next item by modeling recent interactions, but they often struggle to capture long-term preferences or multi-step behaviors. Recent approaches forecast user behavior trajectories over multiple time steps or conversion stages. Graph Multi-Scale Pyramid Networks (GMPN) [20] capture multi-resolution temporal dynamics and category dependencies for purchase prediction. The Adaptive Intent Transfer Model (AITM) [12] models sequential dependencies in multi-stage conversion funnels via adaptive feature transfer. PinnerFormer [13] employs a Transformer-based model to forecast comprehensive user engagements.

**Diffusion Models in User Behavior Prediction.** Unlike traditional generative models such as VAEs and GANs, diffusion models [21] employ a denoising-based generation process that progressively reverses a multi-step noising procedure. This approach enables more precise alignment between generated samples and the underlying training data distribution and show promising results in various generative tasks. Recently, diffusion models have emerged as the prevailing generative predictor in many user behavior modeling tasks [22, 23, 24]. DiffRec [25] proposes the application of diffusion processes to users' interaction vectors (i.e., multi-hot vectors) to facilitate collaborative recommendation. DiffuRec [15] uses a Transformer backbone to reconstruct target representations based on the noised user's historical interaction behaviors. DCDR [26] adopts a discrete diffusion framework for progressive re-ranking. DreamRec [14] generates oracle next-item embeddings based on user preferences but faces scalability challenges due to the absence of negative sampling. PreferDiff [16] reformulates the BPR loss for diffusion models. DCRec [27] further enhances alignment between generated predictions and user preferences through implicit and explicit conditioning mechanisms. Although promising, these methods are not ideally suited for trajectory prediction due to their position-wised optimization objectives and their disregard for the causal relationships within the predicted trajectory.

# 3 Preliminary

In this section, we introduce notation and formally define the tasks studied in this paper.

**Definition 1** (User Interaction History). Let $\mathcal{I} = \{1, 2, \ldots, M\}$ denote the item universe of size $M$. For each user $u$, we denote their observed interaction history up to time $n$ as:

$$H_u = (i_{u,1}, i_{u,2}, \ldots, i_{u,n}) \in \mathcal{I}, \tag{1}$$

where $i_{u,j}$ is the $j$-th item interacted by user $u$ and $n$ is the history length.

**Definition 2** (Multi-step Top-$K$ Sequence Prediction). Given a user history $H_u = (i_{u,1}, \ldots, i_{u,n})$, the goal is to predict the next $k$ items:

$$S_u = (i_{u,n+1}, i_{u,n+2}, \ldots, i_{u,n+k}), \tag{2}$$

in which for each future position $j \in \{1, \ldots, k\}$, the model outputs a ranked list of $K$ candidates

$$A_{u,j} = (a_{u,j,1}, a_{u,j,2}, \ldots, a_{u,j,K}), \tag{3}$$

such that $A_{u,j} \subseteq \mathcal{I}$ and $|A_{u,j}| = K$. The evaluation metrics are based on the inclusion of the ground-truth item $i_{u,n+j}$ in $A_{u,j}$.

**Objective of Diffusion Models.** We first revisit the vanilla diffusion models, established by DDPM [21] and introduce the diffusion objective employed in user behavior modeling. The objective optimization of diffusion parameterized by $\theta$ is to model the distribution of user behaviors, denoted by $p_\theta(z_0)$, where $z_0$ is the target sample. In the context of diffusion-based recommendation, $z_0$ corresponds to the embedding of the target item $i$.

To learn the $p_\theta(z_0)$, DDPM first progressively add noise to sample $z_0$ as $\{z_0, \ldots, z_{T-1}, z_T\}$, where $T$ is the number of diffusion steps and $z_T$ is a standard Gaussian noise. Each step can be formulated as $q(z_t|z_{t-1}) = \mathcal{N}(z_t; \sqrt{\alpha_t}z_{t-1}, (1 - \alpha_t)\mathbf{I})$, where the Gaussian parameters $\alpha_t$ varies over time step $t$. This is called the forward process.

The reverse process is recovering the $z_0$ by $\{z_T, \ldots, z_1, z_0\}$, where each denoising step can be also formulated as a Gaussian transition $p_\theta(z_{t-1}|z_t) = \mathcal{N}(z_{t-1}; \mu_\theta(z_t, t), \Sigma_\theta(z_t, t))$. Based on the forward and reversed process, the objective optimization based on ELBO can be formulated as:

$$\mathcal{L}_{\text{ELBO}} = \mathbb{E}_{q_\phi(z_{0:T}|i)} \left[ \underbrace{\sum_{t=1}^{T} C_t ||f_\theta(z_t, t) - z_0||^2}_{\mathcal{L}_{\text{Reconstruction}}} + \underbrace{\log \frac{q_\phi(z_0|i)}{p_\phi(i|z_0)}}_{\mathcal{L}_{\text{Ranking}}} + \underbrace{\log \frac{q(z_T|z_0)}{p_\theta(z_T)}}_{\mathcal{L}_{\text{Regularization}}} \right], \tag{4}$$

where $f_\theta$ is the denoising model and $\phi$ represents the parameters of embedding layers in the recommendation task. It is worth noting that $\phi$ and $\mathcal{L}_i$ is not included in DDPM as it was originally designed for image domains without embedding layers. We provide a detailed derivation in Appendix B.

**Diffusion Model Inference and Behavior Modeling.** Starting from pure Gaussian noise, diffusion models employ the denoising network $f_\theta$ to iteratively generate latent embeddings, culminating in the inferred behavior prediction embedding $z_0$. This embedding $z_0$ is then mapped back to the corresponding item space. In sequential recommendation tasks, the Top-$K$ items from the candidate set are identified as potential user preferences.

# 4 Listwise Preference Diffusion Optimization

In this section, we present **Listwise Preference Diffusion Optimization (LPDO)**, a novel framework for the UBTP task. LPDO departs from standard diffusion-based recommenders by directly instilling listwise preference information into a non-autoregressive diffusion model in continuous latent space, coupled with a Plackett–Luce ranking objective. This unified generative ranking design raises several nontrivial challenges: most notably, how to bridge the gap between continuous denoising and discrete Top-$K$ list creation, how to derive a tractable variational bound that jointly accommodates reconstruction and listwise likelihood, and how to balance generation fidelity against preference alignment without destabilizing training. In the remainder of this section, we describe how we overcome these hurdles through (i) an end-to-end differentiable scoring head mapping denoised latents to item logits, (ii) the preference causal transformer, which models the directional dependencies among

denoised item representations to capture causal preference order and enhance listwise consistency and (iii) a tight ELBO decomposition that seamlessly integrates Plackett–Luce terms.

Our approach leverages a non-autoregressive diffusion model in continuous latent space, combined with a Plackett-Luce ranking objective. We describe the key components below.

## 4.1 Diffusion Framework for UBTP

**Objective of Diffusion UBTP.** We extend the diffusion optimization framework for user trajectory prediction based on DCRec [27]. Unlike prior works that add noise solely to the target item embedding $\mathbf{z}_0$ [14, 15, 16], DCRec concatenates the target item embedding with the corresponding history embedding, thereby introducing an explicit conditioning signal through $\mathbf{z}_0$. This concatenation strategy has also been shown to be an effective diffusion paradigm in NLP tasks [28]. Motivated by this, we adopt a similar design for the UBTP task by concatenating the target trajectory with its corresponding historical context. Let $\mathbf{z}_0 = (z_{0,1}, \ldots, z_{0,k}) \in \mathbb{R}^{k \times d}$ be the latent embeddings corresponding to the future $k$ items. The concatenated sample can be formulated as $\mathbf{X_0} = \text{concat}(\mathcal{H}, \mathbf{z}_0)$. Therefore, as illustrated in Figure 2, we reformulate the forward and reverse processes as $q(\mathbf{X}_t|\mathbf{X}_{t-1})$ and $p_\theta(\mathbf{X}_{t-1}|\mathbf{X}_t, \mathcal{H})$, respectively, based on which Equation (4) can be extended as follows:

$$\mathcal{L}_{\text{UBTP}} = \mathbb{E}_{q_\phi(\mathbf{X}_{0:T}|S_u)} \left[ \underbrace{\sum_{t=1}^{T} C_t ||f_\theta(\mathbf{X}_t, t) - \mathbf{X}_0||^2}_{\mathcal{L}_{\text{Reconstruction}}} + \underbrace{\log \frac{q_\phi(\mathbf{X}_0|S_u)}{p_\phi(S_u|\mathbf{X}_0)}}_{\mathcal{L}_{\text{Ranking}}} + \underbrace{\log \frac{q(\mathbf{X}_T|\mathbf{X}_0)}{p_\theta(\mathbf{X}_T)}}_{\mathcal{L}_{\text{Regularization}}} \right], \quad (5)$$

where $S_u$ is predicted item lists corresponding to $\mathbf{z}_0$.

**Preference Causal Transformer.** Unlike the next-item prediction target in previous user behavior tasks, trajectory prediction exhibits strong causality and a clear evolution of user preferences within the trajectory. Therefore, as shown in Figure 2 bottom, we employ a causal transformer backbone as $f_\theta$. The concatenated embedding $\mathbf{X}_t$ and time embedding $t$ are sent to the causal attention layer. Following [27], the clean history embedding $\mathcal{H}_0$ is then added as a conditioning signal through a cross-attention layer. This model effectively captures causal dependencies and user preferences while denoising the input, thereby enhancing the quality of the generated outputs $\mathbf{z}$.

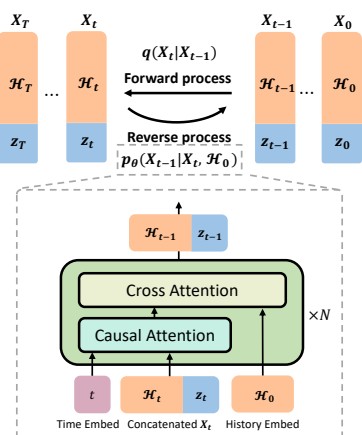

Figure 2: Illustration of LPDO.

## 4.2 Bridging Diffusion and Sequential Listwise Ranking

To ground our sequence-level diffusion objective in established learning-to-rank theory, we begin by recalling the classical Bayesian Personalized Ranking (BPR) loss [29], which optimizes pairwise preferences. Given a user $u$, a positive item $i^+$ and a negative item $i^-$, BPR minimizes:

$$\mathcal{L}_{\text{BPR}} = -\ln \sigma(\hat{y}_{u,i^+} - \hat{y}_{u,i^-}), \quad (6)$$

where $\hat{y}_{u,i}$ is the predicted score for user $u$ on item $i$, and $\sigma(\cdot)$ is the sigmoid function. While effective for one-step ranking, BPR has two major drawbacks in our UBTP task: (i) *Pairwise myopia*: it only compares one positive/negative pair, ignoring interactions among the full future sequence $(i_{u,n+1}, \ldots, i_{u,n+k})$. (ii) *Latent-variable mismatch*: BPR's pointwise score differences cannot readily incorporate the diffusion latent variables $\mathbf{X}_t$ that model an entire trajectory at each step.

To capture full-sequence structure, we turn to the listwise likelihood framework. Let $\pi = [\pi_1, \ldots, \pi_K]$ be a permutation of the $K$ candidate items at one future position $j$. ListMLE [30] maximizes:

$$\mathcal{L}_{\text{ListMLE}} = -\sum_{r=1}^{K} \log \frac{\exp(s_{\pi_r})}{\sum_{i \in \mathcal{C}_r} \exp(s_i)}, \quad (7)$$

$$\text{where} \quad \mathcal{C}_r = \mathcal{C} \setminus \{\pi_1, \ldots, \pi_{r-1}\}, \quad (8)$$

where $s_i$ is the score for item $i$, and $\mathcal{C}$ denotes the candidate set at this prediction step. This listwise objective jointly enforces consistency among all ranks within the position $j$. However, the strict

exclusion of lower-ranked items can produce overly peaked distributions and sparse gradients that impede stable optimization. Thus, we further loosen ListMLE:

$$\mathcal{L}_{\text{Soft-ListMLE}} = -\sum_{r=1}^{K} \log \frac{\exp(s_{\pi_r})}{(1-\gamma)\sum_{i \in \mathcal{C}_r} \exp(s_i) + \gamma \sum_{i \in \mathcal{C}} \exp(s_i)}, \tag{9}$$

where $\gamma \in [0, 1]$ is a penalty factor. Building on $\mathcal{L}_{\text{Soft-ListMLE}}$, we embed listwise ranking into our diffusion-based user trajectory model. Recall from Section 4.1 that we already have:

$$\mathbf{X}_t = \text{concat}\big(\mathcal{H}, (z_{t,1}, \ldots, z_{t,k})\big), \tag{10}$$

We then define a *listwise diffusion loss* over the ground-truth next-item sequence $S_u$:

$$\mathcal{L}_{\text{ListPref}} = -\sum_{j=1}^{k} \ln p_\theta\big(i_{u,n+j} \mid \mathbf{X}_t, \mathcal{H}; \gamma\big), \tag{11}$$

where $p_\theta\big(i_{u,n+j} \mid \mathbf{X}_t, \mathcal{H}; \gamma\big)$ is the Plackett-Luce model when $\gamma = 0$. In this way, $L_{\text{ListPref}}$ inherits the global ranking advantages of ListMLE while fully integrating the diffusion latent dynamics that generate multi-step user trajectories.

### 4.3 Derivation of ListPref Loss

To optimize the joint likelihood of the latent trajectories and the Top-$K$ lists, we consider optimizing Equation (11). This objective is directly intractable because it requires marginalizing over the full diffusion chain $\mathbf{X}_{0:T}$. We introduce the forward noising process $q(\mathbf{X}_{0:T} \mid \mathbf{X}_0)$ as a variational posterior [21] and obtain:

$$\begin{aligned}
\mathcal{L}_{\text{ListPref}} &= -\sum_{j=1}^{k} \ln \int p_\theta\big(i_{u,n+j}, \mathbf{X}_{0:T} \mid \mathcal{H}; \gamma\big) \, d\mathbf{X}_{0:T} \\
&= -\sum_{j=1}^{k} \ln \int q(\mathbf{X}_{0:T} \mid \mathbf{X}_0) \frac{p(\mathbf{X}_T) \prod_{t=1}^{T} p_\theta(\mathbf{X}_{t-1} \mid \mathbf{X}_t, \mathcal{H}) \, p_\theta\big(i_{u,n+j} \mid \mathbf{X}_0, \mathcal{H}; \gamma\big)}{q(\mathbf{X}_{0:T} \mid \mathbf{X}_0)} \, d\mathbf{X}_{0:T} \\
&\leq -\mathbb{E}_{q(\mathbf{X}_{0:T} \mid \mathbf{X}_0)}\left[\sum_{j=1}^{k} \ln p_\theta\big(i_{u,n+j} \mid \mathbf{X}_0, \mathcal{H}; \gamma\big)\right] \\
&\quad + \sum_{t=1}^{T} \mathbb{E}_{q(\mathbf{X}_t \mid \mathbf{X}_0)} \text{KL}\big(q(\mathbf{X}_{t-1} \mid \mathbf{X}_t, \mathbf{X}_0) \, \| \, p_\theta(\mathbf{X}_{t-1} \mid \mathbf{X}_t, \mathcal{H}; \gamma)\big) \; + \; \text{KL}\big(q(\mathbf{X}_T \mid \mathbf{X}_0) \, \| \, p(\mathbf{X}_T)\big).
\end{aligned} \tag{12}$$

### 4.4 Connecting Diffusion Models With Listwise Maximum Likelihood Estimation

#### 4.4.1 Joint Likelihood and ELBO

To formally connect diffusion-based generation with listwise maximum likelihood, we write the joint likelihood of the next-item sequence and the diffusion chain:

$$\sum_{j=1}^{k} \ln p_\theta\big(i_{u,n+j}, \mathbf{X}_{0:T} \mid \mathcal{H}; \gamma\big) = \ln p_\theta(\mathbf{X}_T) + \sum_{t=1}^{T} \ln p_\theta\big(\mathbf{X}_{t-1} \mid \mathbf{X}_t, \mathcal{H}; \gamma\big) + \sum_{j=1}^{k} \ln p_\theta\big(i_{u,n+j} \mid \mathbf{X}_0, \mathcal{H}; \gamma\big), \tag{13}$$

where $p_\theta(\mathbf{X}_{t-1} \mid \mathbf{X}_t, \mathcal{H}; \gamma)$ denotes the reverse diffusion kernel and $p_\theta(i_{u,n+j} \mid \mathbf{X}_0, \mathcal{H}; \gamma)$ models the listwise likelihood of the true next item under the denoised latent state.

Maximizing the marginal likelihood $\sum_{j=1}^{k} \ln p_\theta(i_{u,n+j} \mid \mathcal{H}; \gamma)$ is intractable, as it requires integrating over all diffusion trajectories. To address this, we introduce the variational distribution $q(\mathbf{X}_{0:T} \mid \mathbf{X}_0)$ to approximate

the forward noising process and derive an evidence lower bound (ELBO):

$$\sum_{j=1}^{k} \ln p_\theta\big(i_{u,n+j} \mid \mathcal{H}; \gamma\big) = \ln \int q(\mathbf{X}_{0:T} \mid \mathbf{X}_0) \frac{\sum_{j=1}^{k} p_\theta\big(i_{u,n+j}, \mathbf{X}_{0:T} \mid \mathcal{H}; \gamma\big)}{q(\mathbf{X}_{0:T} \mid \mathbf{X}_0)} \, d\mathbf{X}_{0:T}$$

$$\geq - \sum_{t=1}^{T} \mathrm{KL}\Big( q(\mathbf{X}_{t-1} \mid \mathbf{X}_t, \mathbf{X}_0) \,\big\|\, p_\theta(\mathbf{X}_{t-1} \mid \mathbf{X}_t, \mathcal{H}; \gamma) \Big)$$

$$+ \sum_{j=1}^{k} \mathbb{E}_{q(\mathbf{X}_{0:T} \mid \mathbf{X}_0)}\Big[ \ln p_\theta\big(i_{u,n+j} \mid \mathbf{X}_0, \mathcal{H}; \gamma\big) \Big], \tag{14}$$

The derivation details are given in the Appendix C. Dropping constants and reordering, the final ELBO becomes:

$$\mathcal{L}_{\text{ELBO}} = \sum_{t=1}^{T} \mathrm{KL}\Big( q(\mathbf{X}_{t-1} \mid \mathbf{X}_t, \mathbf{X}_0) \,\big\|\, p_\theta(\mathbf{X}_{t-1} \mid \mathbf{X}_t, \mathcal{H}; \gamma) \Big) - \sum_{j=1}^{k} \mathbb{E}_{q(\mathbf{X}_{0:T} \mid \mathbf{X}_0)}\Big[ \ln p_\theta\big(i_{u,n+j} \mid \mathbf{X}_0, \mathcal{H}; \gamma\big) \Big]. \tag{15}$$

Minimizing $\mathcal{L}_{\text{ELBO}}$ jointly optimizes denoising fidelity (first term) and listwise ranking utility (second term), thus bridging diffusion generation with listwise maximum likelihood estimation. We provide a pseudo-code of the training process in Appendix D. Please refer to it for more details.

## 4.5 Optimization and Inference

**Optimization.** We optimize $\mathcal{L}(\theta)$ via stochastic gradient descent. At inference, we sample from $p_\theta$ by iterative denoising from $\mathbf{z}_T \sim \mathcal{N}(0, I)$ down to $\mathbf{z}_1$, then compute scores $s_{j,i}$ and select Top-$K$ per position in parallel:

$$\mathcal{L}_{\text{LPDO}} = \underbrace{\lambda \mathcal{L}_{\text{Simple}}}_{\text{Reconstruction Loss}} + \underbrace{(1-\lambda)\mathcal{L}_{\text{ListPref}}}_{\text{Learning Preference}} + \underbrace{\mathcal{L}_{\text{Reg}}}_{\text{Regularization Loss}} \tag{16}$$

Here, $\mathcal{L}_{\text{Simple}}$ and $\mathcal{L}_{\text{Reg}}$ are defined in Equation (5). This unified objective jointly enforces trajectory consistency and listwise ranking fidelity during training.

**Inference.** At test time, given a user history $H_u$, we sample a $k$-step future trajectory as follows:

$$\mathbf{X}_T \sim \mathcal{N}\big(0, \mathbf{I}\big), \quad \mathbf{X}_{t-1} = \mu_\theta\big(\mathbf{X}_t, t\big) + \Sigma_\theta\big(\mathbf{X}_t, t\big)^{1/2} \epsilon_t, \quad \epsilon_t \sim \mathcal{N}(0, \mathbf{I}), \quad t = T, \dots, 1. \tag{17}$$

After obtaining the denoised latent $\mathbf{X}_1$, we compute for each future position $j = 1, \dots, k$ the ranking scores:

$$s_{j,i} = e_i^\top \phi_\theta(\mathbf{X}_1)_j, \quad i \in \mathcal{I}, \tag{18}$$

and select the Top-$K$ items:

$$A_{u,j} = \text{TopK}\big(\{s_{j,i}\}_{i \in \mathcal{I}}\big). \tag{19}$$

We also provide a pseudo-code for the inference process in Appendix D. Please refer to it for more details.

## 4.6 Evaluation Metric for UBTP

Commonly used evaluation metrics for next-item prediction tasks, such as HR@K and NDCG@K, are not well-suited for UBTP tasks because they fail to effectively capture the sequential relationships within long-term predictions. Similarly, sequence-based metrics from NLP tasks, such as BLEU [13] and ROUGE [31], are also inappropriate for UBTP tasks due to their emphasis on strict Top-1 matching. This requirement is challenging to meet in user behavior tasks, where data sparsity is a significant issue.

To evaluate the long-term prediction ability, inspired by the conventional Top-$K$ metric HR@K, we use a new list-wise metric SeqMatch@K (SM@K) to measure the similarity of two trajectories as follow:

$$\text{SeqMatch@K} = \frac{1}{|\mathcal{D}_{\text{test}}|} \sum_{s \in \mathcal{D}_{\text{test}}} \mathbb{I}\left( \bigwedge_{j=1}^{k} \big(i_{u,n+j} \in A_{u,j}\big) \right) \tag{20}$$

where $\mathcal{D}_{\text{test}}$ is the test dataset. SeqMatch@K measures the strict consistency of a trajectory prediction model. It calculates the percentage of test trajectories where every target item in the trajectory appears in the model's Top-$K$ predictions for their respective positions. We provides an example of SeqMatch@K in Appendix E.

# 5 Experiments

In this section, we aim to answer the following research questions:

- **RQ1**: How does LPDO perform on the UBTP task compared to state-of-the-art baselines?
- **RQ2**: How does LPDO benefits from $\mathcal{L}_{\text{LPDO}}$ comparing to other diffusion-based methods?
- **RQ3**: What is the impact of different hyperparameters (e.g., balance factors $\lambda$ and $\gamma$) on LPDO's performance?
- **RQ4**: What are the inference cost and model complexity of LPDO compared to the baseline methods?

## 5.1 Training Setup

**Datasets.** We evaluate our approach on three sequential recommendation datasets: Amazon Beauty [32], MovieLens-1M [33], and LastFM [34]. We adopt the standard *Leave-One-Out* data-splitting strategy commonly used in sequential recommendation tasks, where the *"One"* in our setting refers to a user trajectory with a predefined sequence length. The trajectory length is set to 3 for Amazon Beauty, and 5 and 10 for MovieLens-1M and LastFM, respectively. Detailed dataset statistics are provided in Appendix G.

**Baselines.** Since UBTP is a newly proposed task, we conduct a comprehensive comparison of LPDO against seven representative baselines from sequential modeling. These include three conventional methods: FPMC [35], SASRec [18], and STOSA [36]; four diffusion-based generative methods: DiffuRec [15], DreamRec [14], PreferDiff [16] and DCRec [27]. Among them, all baselines are modified from their default training objectives to support UBTP by applying the loss to each predicted item. Additionally, we extend SASRec to support auto-regressive generation, denoted as SASRec-AR. This variant predicts the next item at each step, appends the predicted item to the input sequence, and then performs the subsequent prediction based on the updated history.

**Implementation Details**. For all baselines, we carefully tune their hyperparameters to achieve the best performance on the validation set. For our LPDO, we set $\lambda$=0.1 for all datasets, and $\gamma$=0.0/0.3/0.8 for Beauty, MovieLens-1M and LastFM. We set the number of diffusion timesteps to 50 for training and 1 for inference, and the $\beta$ linearly increases in the range of [1e-4, 0.02]. We set the number of candidates $K = M$ during training. All models are trained and evaluated on an NVIDIA RTX 3090 GPU, and the training stopped after 5 evaluations without improvement to prevent overfitting. More implementation details are described Appendix H.

**Evaluation.** To evaluate the whole predicted trajectory, we modify the HR@K and NDCG@K metric, which is for next-item prediction, to sequence HR (SeqHR) and sequence NDCG (SeqNDCG). SeqHR and SeqNDCG are sequence-level evaluation metrics that compute the geometric mean of position-wise HRs or NDCGs. Furthermore, we introduce SeqMatch (see Section 4.6), enforcing strict consistency evaluation across all time steps in a trajectory. Meanwhile, we also use Perplexity (PPL) [37] to quantify the uncertainty of the predicted trajectory. For numerical stability, PPL is scaled by applying a logarithmic transformation. Following [15, 38], we rank all candidate items for each predicted item in the target trajectory.

## 5.2 Performance Comparison (RQ1)

In this section, Table 1 reports the comparison results between our method and 8 different baseline methods on three datasets with three different trajectory length settings. LPDO demonstrates significant improvements across all datasets, highlighting the effectiveness of our diffusion-based approach for UBTP tasks. By incorporating a listwise preference-aware optimization strategy, our framework facilitates the coherent and accurate generation of user behavior trajectories. This enhancement improves consistency within the diffusion process, which is evidenced by the performance gains over baseline methods.

Moreover, diffusion-based methods with ranking loss, such as DiffuRec and DCRec, generally outperform non-diffusion methods. In contrast, diffusion models without ranking loss, such as DreamRec and PreferDiff, struggle with trajectory prediction tasks, resulting in nearly zero values in all ranking metrics with significantly higher PPL (e.g., PPL of PreferDiff is >500.0). We attribute this performance gap to embedding collapse; further analysis appears in Section J. This trend indicates the advantages of generative modeling techniques and the integration of ranking loss in capturing complex user behaviors and enabling causal future behavior prediction. Among these methods, diffusion-based approaches like DiffuRec and DCRec demonstrate superior performance, likely due to their ability to capture the inherent uncertainty of user interests. Notably, LPDO surpasses both DiffuRec and DCRec across all datasets, showcasing that the incorporation of listwise optimization further enhances the modeling of user preferences in both the forward and reverse diffusion processes.

## 5.3 Benefit From $\mathcal{L}_{\text{LPDO}}$ (RQ2)

In Section 4.4, we discuss how LPDO handles high-ranking items with larger gradients. Empirically, we find that there are two main advantages.

Table 1: Overall performance comparison across different datasets and trajectory length settings.

| Data | Metric | FPMC | SASRec | SASRec-AR | STOSA | DiffuRec | DreamRec | PerferDiff | DCRec | LPDO | Improve. |
|---|---|---|---|---|---|---|---|---|---|---|---|
| Beauty (len=3) | SH@5 | 0.0236 | 0.0219 | 0.0218 | 0.0238 | 0.0274 | 0.0002 | 0.0025 | 0.0250 | **0.0307** | 12.04% |
| | SH@10 | 0.0376 | 0.0466 | 0.0455 | 0.0407 | 0.0442 | 0.0006 | 0.0055 | 0.0388 | **0.0503** | 7.94% |
| | SN@5 | 0.0129 | 0.0114 | 0.0114 | 0.0120 | 0.0149 | 0.0001 | 0.0015 | 0.0135 | **0.0164** | 10.07% |
| | SN@10 | 0.0158 | 0.0199 | 0.0195 | 0.0171 | 0.0188 | 0.0003 | 0.0025 | 0.0159 | **0.0248** | 24.62% |
| | SM@50 | 0.0253 | 0.0257 | 0.0238 | 0.0177 | 0.0363 | 0.0000 | 0.0000 | 0.0432 | **0.0521** | 20.60% |
| | PPL | 37.56 | 51.25 | 52.59 | 54.89 | 36.89 | 89.55 | > 500.0 | 38.90 | **33.86** | 8.21% |
| ML-1M (len=5) | SH@5 | 0.0157 | 0.0895 | 0.0795 | 0.0865 | 0.0894 | 0.0028 | 0.0004 | 0.1107 | **0.1218** | 10.03% |
| | SH@10 | 0.0371 | 0.1365 | 0.1366 | 0.1476 | 0.1540 | 0.0054 | 0.0165 | 0.1823 | **0.1983** | 8.78% |
| | SN@5 | 0.0098 | 0.0434 | 0.0445 | 0.0493 | 0.0506 | 0.0016 | 0.0049 | 0.0621 | **0.0679** | 9.34% |
| | SN@10 | 0.0179 | 0.0573 | 0.0574 | 0.0627 | 0.0649 | 0.0023 | 0.0073 | 0.0759 | **0.0825** | 8.70% |
| | SM@50 | 0.0160 | 0.1045 | 0.1049 | 0.1074 | 0.1168 | 0.0000 | 0.0000 | 0.1458 | **0.1559** | 6.92% |
| | PPL | 38.43 | 33.62 | 33.58 | 33.47 | 32.47 | 165.59 | > 500 | 31.42 | **30.36** | 3.37% |
| ML-1M (len=10) | SH@5 | 0.0058 | 0.0581 | 0.0560 | 0.0476 | 0.0624 | 0.0022 | 0.0077 | 0.0717 | **0.0819** | 14.22% |
| | SH@10 | 0.0140 | 0.1033 | 0.0985 | 0.0864 | 0.1075 | 0.0043 | 0.0101 | 0.1277 | **0.1419** | 11.12% |
| | SN@5 | 0.0036 | 0.0331 | 0.0320 | 0.0276 | 0.0356 | 0.0012 | 0.0043 | 0.0412 | **0.0461** | 11.89% |
| | SN@10 | 0.0068 | 0.0446 | 0.0426 | 0.0377 | 0.0456 | 0.0018 | 0.0006 | 0.0550 | **0.0603** | 9.63% |
| | SM@50 | 0.0000 | 0.0299 | 0.0261 | 0.0214 | 0.0311 | 0.0000 | 0.0000 | 0.0381 | **0.0447** | 17.32% |
| | PPL | 85.55 | 68.99 | 70.41 | 73.92 | 68.35 | 305.19 | > 500 | 68.58 | **67.44** | 1.66% |
| Last-FM (len=5) | SH@5 | 0.1636 | 0.1816 | 0.1829 | 0.1537 | 0.1793 | 0.0433 | 0.0035 | 0.1931 | **0.2507** | 29.83% |
| | SH@10 | 0.2311 | 0.2523 | 0.2564 | 0.2184 | 0.2458 | 0.0531 | 0.0065 | 0.2681 | **0.3244** | 26.52% |
| | SN@5 | 0.0830 | 0.0917 | 0.0908 | 0.0760 | 0.0906 | 0.0205 | 0.0023 | 0.0972 | **0.1260** | 29.63% |
| | SN@10 | 0.0897 | 0.0953 | 0.0972 | 0.0836 | 0.0921 | 0.0185 | 0.0028 | 0.1015 | **0.1195** | 17.73% |
| | SM@50 | 0.1636 | 0.1578 | 0.1636 | 0.1463 | 0.1638 | 0.0189 | 0.0000 | 0.1723 | **0.2357** | 36.81% |
| | PPL | 39.29 | 28.26 | 28.20 | 30.65 | 30.5495 | 149.98 | > 500.0 | 29.28 | **27.28** | 3.26% |
| Last-FM (len=10) | SH@5 | 0.1080 | 0.1163 | 0.1148 | 0.0888 | 0.1209 | 0.0290 | 0.0000 | 0.1311 | **0.1705** | 30.05% |
| | SH@10 | 0.1703 | 0.1606 | 0.1654 | 0.1495 | 0.1715 | 0.0407 | 0.0074 | 0.1881 | **0.2362** | 25.57% |
| | SN@5 | 0.0598 | 0.0608 | 0.0597 | 0.0471 | 0.0625 | 0.0141 | 0.0000 | 0.0673 | **0.0879** | 30.61% |
| | SN@10 | 0.0686 | 0.0610 | 0.0645 | 0.0624 | 0.0665 | 0.0159 | 0.0033 | 0.0730 | **0.0889** | 21.78% |
| | M@50 | 0.0713 | 0.0635 | 0.0566 | 0.0449 | 0.0586 | 0.0098 | 0.0000 | 0.0557 | **0.0762** | 6.87% |
| | PPL | 83.16 | 61.69 | 61.56 | 66.41 | 65.29 | 283.50 | > 500.0 | 61.32 | **60.93** | 0.64% |

SH@K, SN@K, and SM@K denote SeqHR@K, SeqNDCG@K, and SeqMatch@K, respectively. For all metrics except PPL, higher values indicate better performance. **Bold** results indicate the best results, while underlined results denote the second-best. *Improve.* denotes the relative improvement of LPDO over the strongest baseline. The improvements are statistically significant ($p < 0.05$). For detailed performance comparison with more metrics, please refer to Appendix K.

**Faster convergence.** LPDO converges faster than other diffusion-based models employing different ranking losses. As illustrated in Figure 3 (a), LPDO converges within approximately 25 epochs, whereas its counterpart DCRec, which adopts a cross-entropy ranking loss, requires around 33 epochs to reach convergence.

**Higher position-wise performance.** Figure 3 (b) provides a position-wise comparison of SASRec, DiffuRec, and LPDO on trajectory prediction tasks. The x-axis represents the different positions of the trajectory, and the y-axis represents the prediction accurate measured by HR@5. The shaded areas represent the standard deviation. The results clearly show that LPDO consistently achieves higher HR@5 scores across all prediction positions compared to the other methods, indicating its superior ability to capture causal dependencies and the evolution of user preferences over time. Notably, LPDO exhibits a significant performance advantage, particularly at earlier positions, further highlighting its effectiveness in UBTP tasks.

## 5.4 Ablation Study of Hyperparameters in LPDO (RQ3)

**Penalty factor** $\gamma$. In Section 4.2, we discuss the deviation of the objective of LPDO. The hyperparameter $\gamma$ controls the balance between visiting items in the earlier positions of the trajectory and the remaining items in the pool. Figure 3 (c) illustrates the relationship between different $\gamma$ settings and the model performance on MovieLens-1M dataset. The results indicate that LPDO achieves optimal performance when $\gamma = 0.3$, underscoring the importance of capture user's preference evolution.

**Loss ratio** $\lambda$. In Equation (16), hyperparameter $\lambda$ controls the balance between reconstruct the trajectory from noise and preference-aware ranking in LPDO. Figure 3 (d) shows that setting $\lambda = 0.1$ achieves an appropriate balance between these two objectives. We also evaluate the distinct contributions of different loss components $\mathcal{L}_{\text{ListPref}}$, $\mathcal{L}_{\text{Simple}}$, $\mathcal{L}_{\text{Reg}}$ in MovieLens-1M and LastFM, with trajectory length 5. The results are depicted in Table 2 and we can find that: (1) The model collapses completely when removing $\mathcal{L}_{\text{ListPref}}$, achieving zero values in all ranking metrics with significantly higher PPL. This demonstrates the essential role of ranking loss in maintaining basic prediction functionality. (2) Removing $\mathcal{L}_{\text{Simple}}$ causes severe performance degradation, particularly on MovieLens-1M where SeqHR@5 drops by 53.8%, indicating the diffusion reconstruction loss critically stabilizes the learning process. (3) Omitting $\mathcal{L}_{\text{Reg}}$ results in suboptimal performance, indicating that regularization plays a crucial role in the diffusion optimization process.

**Model Backbone**. We further conduct an ablation study on the Transformer backbone used in LPDO. Please refer to Appendix I for detailed results and analysis.

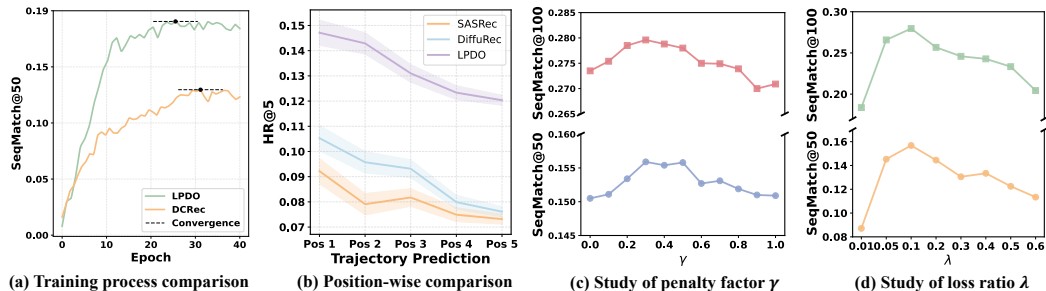

| (a) Training process comparison | (b) Position-wise comparison | (c) Study of penalty factor $\gamma$ | (d) Study of loss ratio $\lambda$ |

Figure 3: Ablation and analysis of LPDO on ML-1M (len=5) dataset. (a) Training process comparison between LPDO and DCRec, showing faster convergence and higher SeqMatch@50 for LPDO. (b) Position-wise comparison of HR@5 across different models, where LPDO consistently outperforms SASRec and DiffuRec at each position of the predicted trajectory. (c) Impact of penalty factor $\gamma$ of $\mathcal{L}_{\text{Total}}$. (d) Effect of loss ratio $\lambda$, indicating the best performance is achieved at moderate values of $\lambda$.

Table 2: Ablation study of different optimization components in LPDO.

| Model | MovieLens-1M (len=5) | | | | LastFM (len=5) | | | |
|---|---|---|---|---|---|---|---|---|
| | SeqHR@5↑ | SeqNDCG@5↑ | SeqMatch@50↑ | PPL↓ | SeqHR@5↑ | SeqNDCG@5↑ | SeqMatch@50↑ | PPL↓ |
| w/o-$\mathcal{L}_{\text{ListPref}}$ | 0.0000 | 0.0000 | 0.0000 | 76.54 | 0.0000 | 0.0000 | 0.0000 | 37.36 |
| w/o-$\mathcal{L}_{\text{Simple}}$ | 0.0563 | 0.0321 | 0.0370 | 34.11 | 0.1452 | 0.0734 | 0.1345 | 29.52 |
| w/o-$\mathcal{L}_{\text{Reg}}$ | 0.0646 | 0.0366 | 0.0634 | 32.99 | 0.2284 | 0.1173 | 0.2123 | 27.87 |
| $\mathcal{L}_{\text{LPDO}}$ | 0.1218 | 0.0678 | 0.1567 | 30.35 | 0.2507 | 0.1260 | 0.2357 | 27.28 |

## 6 Model Inference Cost and Complexity Analysis (RQ4)

In this section, we analyze the inference cost and model complexity of LPDO. We evaluate LPDO against baseline models on the MovieLens-1M dataset with trajectory lengths of 5 and 10, as summarized in Table 3. The inference time for the length-5 setting is generally higher than that for length-10, primarily due to the larger data volume (see Table 5). Notably, the number of denoising steps in LPDO can be flexibly adjusted, allowing the model to remain efficient for real-time trajectory prediction tasks. Although reducing the number of denoising steps may slightly decrease performance, it substantially lowers computational cost. The overall model complexity remains comparable across all methods, as they employ a similar Transformer-based backbone.

Table 3: Comparison of prediction performance with inference cost and complexity.

| Model | MovieLens-1M (len=5) | | | MovieLens-1M (len=10) | | | Complexity |
|---|---|---|---|---|---|---|---|
| | Time Cost↓ | SeqMatch@50↑ | PPL↓ | Time Cost↓ | SeqMatch@50↑ | PPL↓ | |
| SASRec | 3.34s | 0.1047 | 33.62 | 3.29s | 0.0299 | 68.99 | $\mathcal{O}(nd^2 + dn^2)$ |
| SASRec-$_{\text{AR}}$ | 4.01s | 0.1049 | 33.52 | 5.30s | 0.0261 | 70.41 | $\mathcal{O}(nd^2 + dn^2)$ |
| DiffuRec | 24.68s | 0.1168 | 32.47 | 21.51s | 0.0311 | 68.35 | $\mathcal{O}(nd^2 + dn^2)$ |
| DCRec | 12.83s | 0.1458 | 31.42 | 11.18s | 0.0381 | 68.58 | $\mathcal{O}(nd^2 + dn^2)$ |
| LPDO (Step=1) | 3.40s | 0.1559 | 30.36 | 3.34s | 0.0447 | 67.44 | $\mathcal{O}(nd^2 + dn^2)$ |
| LPDO (Step=5) | 4.14s | 0.1569 | 30.36 | 4.07s | 0.0446 | 67.43 | $\mathcal{O}(nd^2 + dn^2)$ |
| LPDO (Step=25) | 8.01s | 0.1575 | 30.35 | 7.65s | 0.0450 | 67.43 | $\mathcal{O}(nd^2 + dn^2)$ |
| LPDO (Step=50) | 13.46s | 0.1575 | 30.35 | 12.26s | 0.0452 | 67.43 | $\mathcal{O}(nd^2 + dn^2)$ |

All results are measured by an NVIDIA RTX 3090 GPU. $d$ and $n$ are the representation dimension and sequence length, respectively.

## 7 Conclusion

In this paper, we introduced the task of User Behavior Trajectory Prediction, which goes beyond next-item recommendation by forecasting coherent, ordered sequences of future user actions. To address the inability of existing diffusion-based predictors to capture global listwise dependencies, we proposed Listwise Preference Diffusion Optimization (LPDO), a novel framework that seamlessly integrates a Plackett–Luce ranking signal into the diffusion ELBO. We derived a tight variational lower bound that couples reconstruction fidelity with listwise ranking likelihood, and we presented SeqMatch, a trajectory-level metric for rigorous multi-step evaluation. Experiments on real-world datasets show that LPDO offers consistent improvement in accuracy, sequence coherence, and uncertainty estimation, with a practical convergence rate. In future work, we plan to incorporate richer context features, extend to longer trajectories, and explore other sequential decision tasks.

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

# Supplementary Material

## A  Limitation

Firstly, LPDO depends on manually tuned hyperparameters—specifically, the loss ratio $\lambda$ and penalty factor $\gamma$, to balance reconstruction and ranking objectives. While we show stable results across moderate values of these parameters (Section 5.1), automated or adaptive tuning strategies could further reduce human effort and improve robustness. Secondly, evaluation focuses on Top-$K$ ranking metrics (SeqHR, SeqNDCG) and strict trajectory matching (SeqMatch). These offline metrics may not fully capture user satisfaction or long-term engagement in live settings. Future work should investigate additional measures, such as personalized utility scores or online A/B tests, to assess real-world impact.

## B  Optimization Deviation of Diffusion Models

### B.1  Diffusion Models for Continuous Domains

A generative model is expected to SeqMatch estimation distribution $p_\theta(z_0)$ to the ground truth distribution $p(z_0)$ as closely as possible, where $\theta$ represents learnable parameters and $z_0 \in \mathbb{R}^d$ is a set of samples. For vanilla diffusion models in continuous domains, they model the sample $z_0$ as a Markov chain with a sequence of latent variables $\{z_T, \ldots, z_1, z_0\}$, where $z_T$ is a standard Gaussian, and each variable is in $\mathbb{R}^d$. By this means, the latent dimension is exactly equal to the data dimension of $z_0$, and the diffusion model progressively recovers $z_0$ from Gaussian noise. Each denoising transition $z_t \to z_{t-1}$ is parameterized by a linear Gaussian transition $p_\theta(z_{t-1}|z_t) = \mathcal{N}(z_{t-1}; \mu_\theta(z_t, t), \Sigma_\theta(z_t, t))$, calculated by a denoising DNN model $f_\theta(z_t, t)$. Therefore, the joint distribution of diffusion models can be written as:

$$p_\theta(z_{0:T}) = p_\theta(z_T) \prod_{t=1}^{T} p_\theta(z_{t-1}|z_t) \tag{21}$$

$$where, p_\theta(z_T) = \mathcal{N}(z_T; \mathbf{0}, \mathbf{I}) \tag{22}$$

To train the denoising model $f_\theta(z_t, t)$, diffusion models first construct a progressive noising process $\{z_0, \ldots, z_{T-1}, z_T\}$, known as the forward process. Corresponding to this, the denoising process is called the reverse process. The forward process adds Gaussian noise to $z_0$ step-by-step until the final latent $z_T$ is guaranteed to be a standard Gaussian. Each noising transition $z_{t-1} \to z_t$ is defined as a linear Gaussian transition $q(z_t|z_{t-1}) = \mathcal{N}(z_t; \sqrt{\beta_t - 1}z_{t-1}, \beta_t\mathbf{I})$, where the Gaussian parameters $\beta_t$ varies over time step $t$. Due to the Markov property and the Bayes rule, the transition can be rewritten with $z_0$ as:

$$q(z_t|z_{t-1}) = q(z_t|z_{t-1}, z_0) = \frac{q(z_{t-1}|z_t, z_0)q(z_t|z_0)}{q(z_{t-1}|z_0)} \tag{23}$$

To optimize the parameter $\theta$, the diffusion model is trained to maximize the log-likelihood of $p_\theta(x)$ according to observed samples $z_0$. This objective can be formulated as $\arg\max_\theta \log p_\theta(z_0)$. Mathematically, we can derive a proxy objective called the Evidence Lower Bound (ELBO), a lower bound of the likelihood term $\log p_\theta(z_0)$. Formally, the equation of the ELBO is:

$$\mathbb{E}_{q(z_{1:T}|z_0)} \left[ \log \frac{p_\theta(z_{0:T})}{q(z_{1:T}|z_0)} \right] \tag{24}$$

Rather than directly maximizing the likelihood term, diffusion models minimize the minus ELBO as:

$$\arg\max_\theta \log p_\theta(z_0) \approx \arg\max_\theta \mathbb{E}_{q(z_{1:T}|z_0)} \left[ \log \frac{p_\theta(z_{0:T})}{q(z_{1:T}|z_0)} \right] \tag{25}$$

$$= \arg\min_\theta \mathbb{E}_{q(z_{1:T}|z_0)} \left[ \log \frac{q(z_{1:T}|z_0)}{p_\theta(z_{0:T})} \right] \tag{26}$$

Combining with Equations (21) and (23), the expectation in Equation (26) can be further derived as:

$$\log \frac{q(z_{1:T}|z_0)}{p_\theta(z_{0:T})} = \log \frac{q(z_T|z_0)}{p_\theta(z_T)} + \sum_{t=2}^{T} \log \frac{q(z_{t-1}|z_t, z_0)}{p_\theta(z_{t-1}|z_t))} - \log p_\theta(z_0|z_1) \tag{27}$$

Therefore, based on Equations (26) and (27), the ELBO-based optimization object of parameter $\theta$ is to minimise

$$\mathcal{L}_{\text{elbo}}(z_0) = \mathbb{E}_{q(z_{1:T}|z_0)} \left[ \log \frac{q(z_T|z_0)}{p_\theta(z_T)} + \sum_{t=2}^{T} \log \frac{q(z_{t-1}|z_t, z_0)}{p_\theta(z_{t-1}|z_t))} - \log p_\theta(z_0|z_1) \right]. \tag{28}$$

To simplify and combine the second and third terms in Equation (28), recent research [39] derivative a simple surrogate objective to obtain a mean-squared error term:

$$\mathcal{L}_{\text{elbo}}(z_0) = \mathbb{E}_{q(z_{1:T}|z_0)} \left[ \log \frac{q(z_T|z_0)}{p_\theta(z_T)} + \sum_{t=1}^{T} C_t ||f_\theta(z_t, t) - z_0||^2 \right] \tag{29}$$

where $C_t$ is constants associated with timesteps $t$.

## B.2 Discrete Diffusion Models for Discrete Domains in Sequential Recommendation

Motivated by diffusion models in text domains [40], we extend continuous diffusion models to discrete item domains. Considering discrete item $z$ from the item pool $\mathcal{Z}$, the Markov chain in the forward and reverse processes are extending as $\{i, z_0, \ldots, z_T\}$ and $\{z_T, \ldots, z_0, i\}$, where $i$ is the discrete item ID. Specifically, to map the discrete variables into continuous domains, we define a learnable embedding function $\text{Emb}(\cdot)$. The forward transition process of $z$ is defined as $q_\phi(z_0|i) = \mathcal{N}(z_0; \text{Emb}(i), \mathbf{0})$, where $\phi$ represents the learnable parameters in $\text{Emb}(i)$. As for the reverse process, we define the predicted distribution of $z$ as $p_\phi(i) = \arg\max_z S(z_0, \text{Emb}(\mathcal{I}))$, where $S$ is the cosine similarity between $z_0$ and each item embedding of $\text{Emb}(\mathcal{I})$. However, the transition distribution $p_\phi(i|z_0)$ is implicit.

Therefore, we extend the ELBO in Equation (26) to include discrete item $i$ as

$$\arg\max_\theta \log p_\theta(i) \approx \arg\min_{\phi,\theta} \mathbb{E}_{q_\phi(z_{0:T}|i)} \left[ \log \frac{q_\phi(z_{0:T}|i)}{p_{\phi,\theta}(i, z_{0:T})} \right] \tag{30}$$

$$\approx \arg\min_{\phi,\theta} \mathbb{E}_{q_\phi(z_{0:T}|i)} \left[ \log \frac{q_\phi(i)q(z_{0:T}|i)}{p_{\phi,\theta}(i, z_{0:T})} \right] \tag{31}$$

And the ELBO-based training objective in Equation (28) is rewritten as

$$\mathcal{L}_{\text{elbo}}(i) = \mathbb{E}_{q_\phi(z_{0:T}|i)} \left[ \log \frac{q(z_T|z_0)}{p_\theta(z_T)} + \sum_{t=1}^{T} C_t ||f_\theta(z_t, t) - z_0||^2 + \log \frac{q_\phi(z_0|i)}{p_\phi(i|z_0)} \right]. \tag{32}$$

There are three terms of objectives in $\mathcal{L}_{\text{elbo}}$: the prior SeqMatching term $\mathcal{L}_T$, the denoising term $\mathcal{L}_t$ and the embedding entropy term $\mathcal{L}_H$.

The prior SeqMatching term $\mathcal{L}_T$ is similar to the same term in Equation (28), which requires the forward process to transit the $z_0$ into pure noise as the $p_\theta(z_T)$ is standard Gaussian. Continuous diffusion models [39] omit this term since there are no associated trainable parameters. It's worth noting that this term is essential in $\mathcal{L}_{\text{elbo}}$ due to the correlation among $i$, $z_0$ and $\phi$. This term is equivalent to minimizing $||z_0||^2$. As the second term $\mathcal{L}_t$ require $f_\theta(z_t, t)$ and $z_0$ as close as possible, we transfer the objective $||z_0||^2$ to $||\sum_{t=1}^{T} f_\theta(z_t, t)||^2$ for fully optimize the DNN models.

The embedding entropy term $\mathcal{L}_H$ can be regarded as a KL-divergence $D_{\text{KL}}(q_\phi(z_0|i)||p_\phi(i|z_0))$. Although $q_\phi(z_0|i)$ is a linear transition, the distribution of $p_\phi(i|z_0)$ is unknown. Therefore, we replace $D_{\text{KL}}(q_\phi(z_0|i)||p_\phi(i|z_0))$ by $D_{\text{KL}}(q(i)||p_\phi(i)) = D_{\text{KL}}(q(i)||p_\phi(i))$. The simplified objective of $\mathcal{L}_{\text{elbo}}$ is:

$$\mathcal{L}_{\text{elbo}}(i) = \mathbb{E}_{q_\phi(z_{0:T}|i)} \left[ \sum_{t=1}^{T} \left( ||f_\theta(z_t, t)||^2 + C_t ||f_\theta(z_t, t) - z_0||^2 \right) \right] + D_{\text{KL}}(q(i)||p_\phi(i)). \tag{33}$$

## B.3 Discrete Diffusion Recommender with Historical Conditioning

In practice, in the sequential recommendation scenario, the predicted target item relies highly on the user's historical interaction $\mathcal{H} = \{h_0, h_1, \ldots, h_n\}$, where $\mathcal{H} \subset \mathcal{I}$. Due to the same item pool $\mathcal{I}$, we use the same embedding $\text{Emb}(i)$ to map historical conditioning into the continuous domains. Recall that the original reverse transition is parameterized as $p_\theta(z_{t-1}|z_t) = \mathcal{N}(z_{t-1}; \mu_\theta(z_t, t), \Sigma_\theta(z_t, t))$. Following the conclusion of previous work [41], we directly incorporate the historical conditioning to $p_\theta(z_{t-1}|z_t)$ as $p_{\phi,\theta}(z_{t-1}|z_t, \mathcal{H}) = \mathcal{N}(z_{t-1}; \mu_{\phi,\theta}(z_t, t, \mathcal{H}), \Sigma_{\phi,\theta}(z_t, t, \mathcal{H}))$, where $\phi$ is the learnable parameters of $\text{Emb}(i)$, and the corresponding denoising model is $f_\theta(z_t, t, \mathcal{H})$.

Therefore, we rewrite Equation (33) with historical conditioning as:

$$\mathcal{L}_{\text{elbo}}^{\text{mix}}(i) = \mathbb{E}_{q_\phi(z_{0:T}|i)} \left[ \sum_{t=1}^{T} \left( ||f_\theta(z_t, t, \mathcal{H})||^2 + C_t ||f_\theta(z_t, t, \mathcal{H}) - z_0||^2 \right) \right] + D_{\text{KL}}(q(i)||p_\phi(i)). \tag{34}$$

## C   Derivation of the ELBO for Diffusion Models with Listwise Maximum Likelihood

To formally connect diffusion-based generation with listwise maximum likelihood, we write the joint likelihood of the next-item sequence and the diffusion chain:

$$\sum_{j=1}^{k} \ln p_\theta\big(i_{u,n+j}, \mathbf{X}_{0:T} \mid \mathcal{H}\big) = \ln p_\theta(\mathbf{X}_T) + \sum_{t=1}^{T} \ln p_\theta\big(\mathbf{X}_{t-1} \mid \mathbf{X}_t,\, \mathcal{H}\big) + \sum_{j=1}^{k} \ln p_\theta\big(i_{u,n+j} \mid \mathbf{X}_0,\, \mathcal{H}\big). \tag{35}$$

Maximizing the marginal likelihood $\sum_{j=1}^{k} \ln p_\theta(i_{u,n+j} \mid \mathcal{H})$ is intractable due to the integration over all diffusion trajectories. To address this, we introduce the variational distribution $q(\mathbf{X}_{0:T} \mid \mathbf{X}_0)$ and derive an evidence lower bound (ELBO):

$$
\begin{aligned}
\ln p_\theta\big(i_{u,n+1:n+k} \mid \mathcal{H}\big) &= \ln \int q(\mathbf{X}_{0:T} \mid \mathbf{X}_0)\, \frac{p_\theta\big(i_{u,n+1:n+k}, \mathbf{X}_{0:T} \mid \mathcal{H}\big)}{q(\mathbf{X}_{0:T} \mid \mathbf{X}_0)}\, d\mathbf{X}_{0:T} \\
&\geq \mathbb{E}_{q(\mathbf{X}_{0:T} \mid \mathbf{X}_0)} \Bigg[ \underbrace{\ln p_\theta(\mathbf{X}_T)}_{\text{const.}} + \sum_{t=1}^{T} \ln p_\theta\big(\mathbf{X}_{t-1} \mid \mathbf{X}_t,\, \mathcal{H}\big) \\
&\quad + \sum_{j=1}^{k} \ln p_\theta\big(i_{u,n+j} \mid \mathbf{X}_0,\, \mathcal{H}\big) - \ln q(\mathbf{X}_{0:T} \mid \mathbf{X}_0) \Bigg] \\
&= - \sum_{t=1}^{T} \mathrm{KL}\Big( q(\mathbf{X}_{t-1} \mid \mathbf{X}_t, \mathbf{X}_0) \,\big\|\, p_\theta(\mathbf{X}_{t-1} \mid \mathbf{X}_t,\, \mathcal{H}) \Big) \\
&\quad + \sum_{j=1}^{k} \mathbb{E}_{q(\mathbf{X}_{0:T} \mid \mathbf{X}_0)} \Big[ \ln p_\theta\big(i_{u,n+j} \mid \mathbf{X}_0,\, \mathcal{H}\big) \Big]. 
\end{aligned} \tag{36}
$$

Dropping constants and reordering, the final ELBO becomes:

$$
\begin{aligned}
\mathcal{L}_{\mathrm{ELBO}} &= \sum_{t=1}^{T} \mathrm{KL}\Big( q(\mathbf{X}_{t-1} \mid \mathbf{X}_t, \mathbf{X}_0) \,\big\|\, p_\theta(\mathbf{X}_{t-1} \mid \mathbf{X}_t,\, \mathcal{H}) \Big) \\
&\quad - \sum_{j=1}^{k} \mathbb{E}_{q(\mathbf{X}_{0:T} \mid \mathbf{X}_0)} \Big[ \ln p_\theta\big(i_{u,n+j} \mid \mathbf{X}_0,\, \mathcal{H}\big) \Big].
\end{aligned} \tag{37}
$$

Minimizing $\mathcal{L}_{\mathrm{ELBO}}$ jointly optimizes denoising fidelity (first term) and listwise ranking utility (second term), thus bridging diffusion generation with listwise maximum likelihood estimation.

## D   Algorithm

In this section, we provide the pseudo-code of the training and inference process in LPDO .

---
**Algorithm 1 Training**

---
**Require:** User historical embedding $\mathcal{H}$, target trajectory embedding $\mathbf{z_0}$, number of denoising steps $T$, trainable network parameters $\theta$ and embedding parameters $\phi$, learning rate $\eta$

1: **repeat**
2:      $t \sim \mathcal{U}(1,T),\, \epsilon \sim \mathcal{N}(\mathbf{0}, \mathbf{I})$                                  ▷ Sample denoising step and noise.
3:      $\mathbf{X}_0 \leftarrow \mathrm{Concat}(\mathcal{H}, \mathbf{z_0})$                                  ▷ Obtain the concatenated sequence.
4:      $\mathbf{X}_t \leftarrow \sqrt{\bar{\alpha}_t}\mathbf{X}_0 + \sqrt{1 - \bar{\alpha}_t}\epsilon$                                  ▷ Adding noise to $\mathbf{X}_0$
5:      $\theta \leftarrow \theta - \eta \nabla_\theta \mathcal{L}_{\mathrm{LPDO}}(\mathbf{X}_t, t, \mathcal{H}; \theta, \phi)$                                  ▷ Gradient descent update $\theta$.
6:      $\phi \leftarrow \phi - \eta \nabla_\phi \mathcal{L}_{\mathrm{LPDO}}(\mathbf{X}_t, t, \mathcal{H}; \theta, \phi)$                                  ▷ Gradient descent update $\phi$.
7: **until** convergence
8: **return** optimized $\theta, \phi$

---

**Algorithm 2 Inference**

**Require:** User historical embedding $\mathcal{H}$, number of denoising steps $T$, denoising model $f_\theta$, target trajectory k, mapping function $p_\phi$

1: $\mathbf{X}_T \sim \mathcal{N}(\mathbf{0}, \mathbf{I})$       ▷ Initialize the model input from Gaussian Noise.
2: **for** $t \leftarrow T \dots 1$ **do**       ▷ Denoise over $T$ steps
3:     $\epsilon \leftarrow \mathcal{N}(0, \mathbf{I})$ if $t > 1$, else 0       ▷ Sample noise if not final step.
4:     $\hat{\mathbf{X}}_0 \leftarrow \frac{1}{\sqrt{\alpha_t}} \mathbf{X}_t - \frac{1-\alpha_t}{\sqrt{1-\bar{\alpha}_t}\sqrt{\alpha_t}} f_\theta(\mathbf{X}_t, t)$       ▷ Denoising.
5:     $\mathbf{X}_{t-1} \leftarrow \sqrt{\bar{\alpha}_{t-1}}\mathbf{X}_0 + \sqrt{1-\bar{\alpha}_{t-1}}\epsilon$       ▷ Update next step.
6: **end for**
7: $\mathbf{z}_0 \leftarrow \text{Extract}(\hat{\mathbf{X}}_0, k)$       ▷ Extract the last $k$ item as prediction.
8: $S_u \leftarrow p_\phi(S_u|\mathbf{z}_0)$       ▷ Mapping from embedding sequence to trajectory.
9: **return** $S_u$

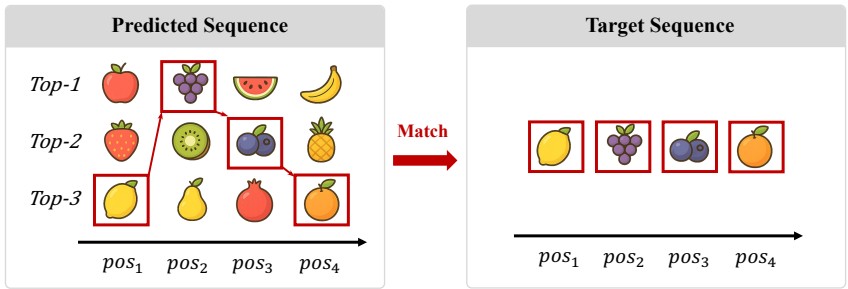

Figure 4: Illustration of the proposed SeqMatch@N metric. The sequence length is 4 and N = 3. The icons were generated using OpenAI's ChatGPT. These icons are solely for illustrative purposes.

## E    Example of SeqMatch@N

Figure 4 provides an example demonstrating how SeqMatch@N evaluates the similarity between predicted and target sequences in the UBTP task. The icons were generated using OpenAI's ChatGPT. These icons are solely for illustrative purposes. At each timestep, the recommender produces k ranked candidate items, forming a Top-$N$ matrix. SeqMatch@N checks whether a valid trajectory exists that selects one candidate per position to match the target sequence. Unlike traditional item-level accuracy (e.g., HR@N, NDCG@N), SeqMatch@N measures sequence-level consistency by capturing both order and positional alignment.

## F    Case Study

Figure 5 illustrates a case study of user trajectory prediction on MovieLens-1M. On the one hand, users may not be interested in certain types of movies. On the other hand, users will also show a decline in interest in the movies they have watched recently. Preference-aware prediction can provide users with movie predictions that better conform to their preferences.

## G    Statistics of Datasets

In this section, we provide the statistics of the datasets used in this study, as shown in Table 4. Empirically, we follow the Leave-One-Out rule to split the dataset. Specifically, for a trajectory length $k$, only sequences in the original dataset with lengths greater than $1 + 3 \times k$ are considered valid. Table 5 reports the statistics of datasets after processing. The number of items is remaining.

Table 4: Statistics of Datasets.

| Dataset | # Sequence | # Items | # Actions | Average Length |
|---|---|---|---|---|
| Beauty | 22,363 | 12,101 | 198,502 | 8.53 |
| MovieLens-1M | 6,040 | 3,416 | 999,611 | 165.50 |
| LastFM | 891 | 1,000 | 1,293,103 | 1254.49 |

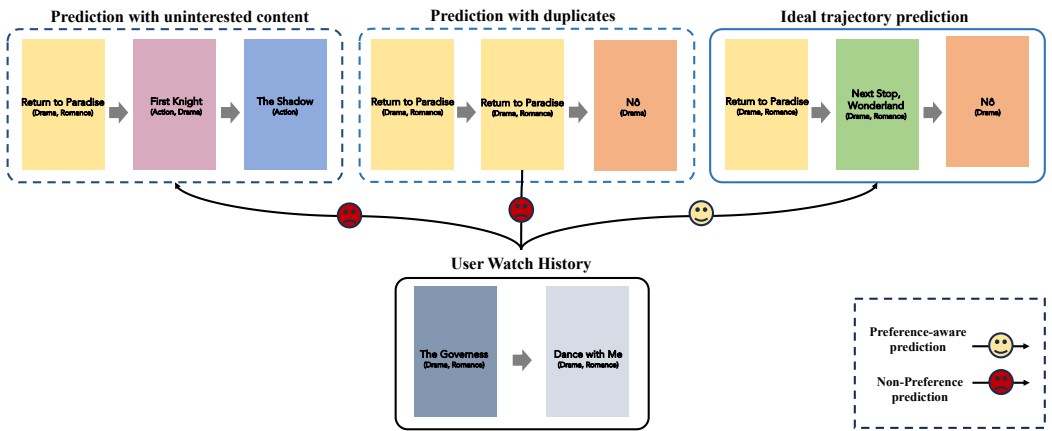

Figure 5: Illustration of preference-aware and non-preference prediction on MovieLens-1M dataset. Due to copyright considerations, we do not show the original movie posters.

Table 5: Statistics of Datasets after Processing.

| Dataset | Trajectory Length | # Sequence(Before) | # Sequence (After) |
|---|---|---|---|
| Beauty | 3 | 22,363 | 1927 |
| MovieLens-1M | 5 | 6,040 | 6040 |
| MovieLens-1M | 10 | 6,040 | 5231 |
| LastFM | 5 | 891 | 829 |
| LastFM | 10 | 891 | 784 |

## H Implementation Details

We implement all models in PyTorch and adopt consistent training configurations across baselines to ensure fair comparison. For both the baselines and our proposed LPDO, we fix the embedding dimension to 128 and the dropout rate to 0.1. For sequential models, the maximum sequence length is fixed at 50. All models are trained using the Adam optimizer [42] with a batch size of 256 and a learning rate selected from $\{0.01, 0.005, 0.001\}$ based on validation performance. For the transformer-based model, the number of blocks is set to 4. Model training is conducted for up to 1,000 epochs with early stopping after 5 epochs of no improvement. To adapt all models to the UBTP task (i.e., next-$k$ item prediction), we modify the output layer to predict multiple future items rather than only the immediate next one. Specifically, their training objective is reformulated from single-item prediction to trajectory-level prediction.

Except for the aforementioned configurations, models with public implementations by previous works, including SASRec [18], STOSA [36], DiffuRec [15], DreamRec [14], and PreferDiff [16], are trained following their respective default settings. For FPMC, we adopt the implementation from ReChorus [43]. For DCRec [27], we reproduce the model and set its loss balance factor to $\lambda = 0.1$.

## I Ablation Study of the Transformer Backbone

In this section, we further analyze the impact of different Transformer backbones on the performance of LPDO. s presented in Table 6, we compare three variants: the bidirectional Transformer [44], the prefix Transformer [45], and the causal Transformer [37]. The results demonstrate that the causal Transformer achieves the best performance, suggesting that its architecture aligns more effectively with the causality nature of the UBTP task.

Table 6: Model performance on MoviesLens-1M (target length=5).

| Backbone | SHR@5 ↑ | SNDCG@5 ↑ | SHR@10 ↑ | SNDCG@10 ↑ | SMatch@50 ↑ | SMatch@100 ↑ | PPL ↓ |
|---|---|---|---|---|---|---|---|
| Bidirectional | 0.1159 | 0.0662 | 0.1925 | 0.0806 | 0.1500 | 0.2709 | 31.35 |
| Prefix | 0.1175 | 0.6517 | 0.1927 | 0.7966 | 0.1521 | 0.2740 | 30.86 |
| Causal | 0.1218 | 0.0679 | 0.1983 | 0.0825 | 0.1559 | 0.2796 | 30.36 |

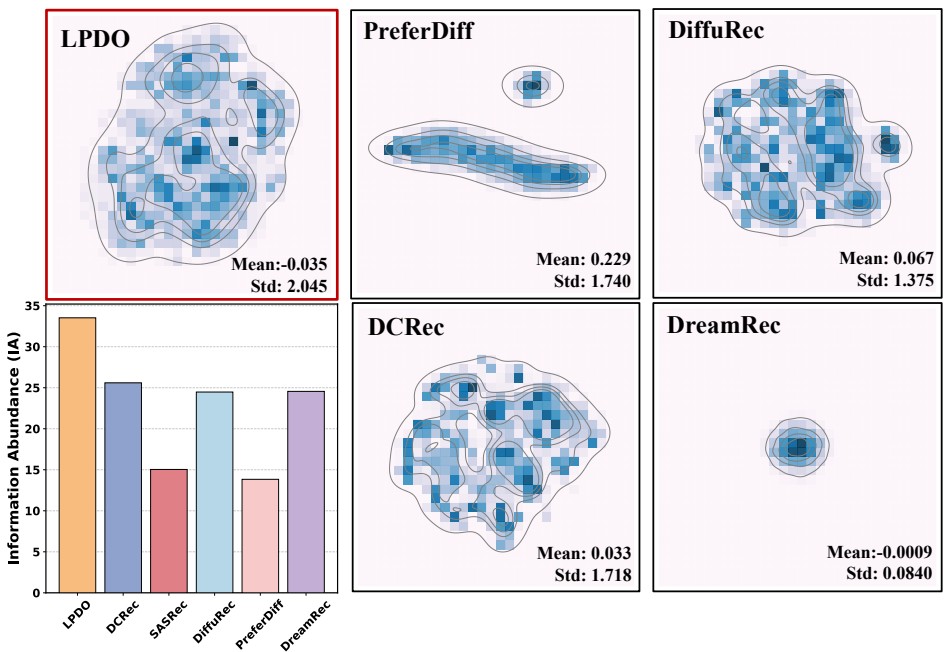

Figure 6: Information Abundance (IA) of model embeddings and T-SNE results of ML-1M (len=5).

## J    Embedding Analysis

In Figure 6, we illustrate the Information Abundance (IA) [46] results of embeddings generated by six models and provide a detailed visualization of the embedding layers for five diffusion-based models. LPDO achieves the highest IA score and exhibits more diverse embeddings, highlighting the effectiveness of our proposed approach. On the other hand, PreferDiff and DreamRec suffer from embedding collapse, likely due to optimization without incorporating ranking loss.

## K    Overall Performance Comparison

In this section, we provide all the evaluation results on Amazon Beauty (len=3), MovieLens (len=5/10) and LastFM (len=5/10) as shown in Table 7, Table 8, Table 9, Table 10 and Table 11, respectively.

Table 7: Overall Performance Comparison on Amazon Beauty (len=3).

| Metric | FPMC | SASRec | SASRec-ar | STOSA | DiffuRec | DreamRec | PreferDiff | DCRec | **LPDO** | Improve. |
|---|---|---|---|---|---|---|---|---|---|---|
| MeanHR@5 | 0.0251 | 0.0226 | 0.0225 | 0.0251 | 0.0303 | 0.0002 | 0.0028 | 0.0260 | **0.0344** | 13.53% |
| MeanHR@10 | 0.0390 | 0.0470 | 0.0458 | 0.0410 | 0.0471 | 0.0007 | 0.0057 | 0.0487 | **0.0549** | 16.56% |
| MeanHR@20 | 0.0622 | 0.0690 | 0.0688 | 0.0583 | 0.0847 | 0.0016 | 0.0096 | 0.0742 | **0.1008** | 19.01% |
| MeanNDCG@5 | 0.0138 | 0.0118 | 0.0118 | 0.0125 | 0.0150 | 0.0001 | 0.0011 | 0.0141 | **0.0184** | 22.67% |
| MeanNDCG@10 | 0.0162 | 0.0201 | 0.0196 | 0.0172 | 0.0192 | 0.0003 | 0.0028 | 0.0169 | **0.0224** | 11.44% |
| MeanNDCG@20 | 0.0193 | 0.0213 | 0.0211 | 0.0183 | 0.0271 | 0.0006 | 0.0031 | 0.0229 | **0.0326** | 20.30% |
| SeqHR@5 | 0.0236 | 0.0219 | 0.0218 | 0.0238 | 0.0274 | 0.0002 | 0.0025 | 0.0250 | **0.0307** | 12.04% |
| SeqHR@10 | 0.0376 | 0.0466 | 0.0455 | 0.0407 | 0.0442 | 0.0006 | 0.0055 | 0.0388 | **0.0503** | 7.94% |
| SeqHR@20 | 0.0619 | 0.0690 | 0.0667 | 0.0581 | 0.0842 | 0.0014 | 0.0093 | 0.0729 | **0.0982** | 16.63% |
| SeqNDCG@5 | 0.0129 | 0.0114 | 0.0114 | 0.0120 | 0.0149 | 0.0001 | 0.0015 | 0.0135 | **0.0164** | 10.07% |
| SeqNDCG@10 | 0.0158 | 0.0199 | 0.0195 | 0.0171 | 0.0188 | 0.0003 | 0.0025 | 0.0159 | **0.0248** | 24.62% |
| SeqNDCG@20 | 0.0193 | 0.0212 | 0.0210 | 0.0182 | 0.0270 | 0.0005 | 0.0030 | 0.0225 | **0.0322** | 19.26% |
| SeqMatch@50 | 0.0253 | 0.0257 | 0.0238 | 0.0177 | 0.0363 | 0.0000 | 0.0000 | 0.0432 | **0.0521** | 20.60% |
| SeqMatch@100 | 0.0598 | 0.0437 | 0.0446 | 0.0318 | 0.0852 | 0.0000 | 0.0004 | 0.0807 | **0.0940** | 10.33% |
| PPL $\downarrow$ | 37.56 | 51.25 | 52.59 | 54.89 | 36.89 | 89.55 | > 500.0 | 38.90 | **33.86** | 8.21% |

Table 8: Overall Performance Comparison on LastFM (len=5).

| Metric | FPMC | SASRec | SASRec-ar | STOSA | DiffuRec | DreamRec | PreferDiff | DCRec | LPDO | Improve. |
|---|---|---|---|---|---|---|---|---|---|---|
| MeanHR@5 | 0.1637 | 0.1819 | 0.1831 | 0.1548 | 0.1797 | 0.0435 | 0.0037 | 0.1977 | **0.2553** | 29.14% |
| MeanHR@10 | 0.2312 | 0.2528 | 0.2560 | 0.2189 | 0.2462 | 0.0536 | 0.0069 | 0.2735 | **0.3295** | 20.48% |
| MeanHR@20 | 0.3313 | 0.3437 | 0.3463 | 0.3019 | 0.3293 | 0.0789 | 0.0144 | 0.3423 | **0.4154** | 19.95% |
| MeanNDCG@5 | 0.0830 | 0.0918 | 0.0909 | 0.0765 | 0.0909 | 0.0206 | 0.0024 | 0.0992 | **0.1277** | 28.73% |
| MeanNDCG@10 | 0.0898 | 0.0954 | 0.0975 | 0.0838 | 0.0922 | 0.0188 | 0.0031 | 0.1033 | **0.1210** | 24.10% |
| MeanNDCG@20 | 0.0997 | 0.1021 | 0.1016 | 0.0897 | 0.0940 | 0.0209 | 0.0046 | 0.1142 | **0.1340** | 17.34% |
| SeqHR@5 | 0.1636 | 0.1816 | 0.1829 | 0.1537 | 0.1793 | 0.0433 | 0.0035 | 0.1931 | **0.2507** | 29.83% |
| SeqHR@10 | 0.2311 | 0.2523 | 0.2564 | 0.2184 | 0.2458 | 0.0531 | 0.0065 | 0.2681 | **0.3244** | 26.52% |
| SeqHR@20 | 0.3310 | 0.3436 | 0.3461 | 0.3016 | 0.3288 | 0.0738 | 0.0140 | 0.3380 | **0.4123** | 19.13% |
| SeqNDCG@5 | 0.0830 | 0.0917 | 0.0908 | 0.0760 | 0.0906 | 0.0205 | 0.0023 | 0.0972 | **0.1260** | 29.63% |
| SeqNDCG@10 | 0.0897 | 0.0953 | 0.0972 | 0.0836 | 0.0921 | 0.0185 | 0.0028 | 0.1015 | **0.1195** | 17.73% |
| SeqNDCG@20 | 0.0996 | 0.1020 | 0.1016 | 0.0896 | 0.0938 | 0.0209 | 0.0043 | 0.1132 | **0.1235** | 9.10% |
| SeqMatch@50 | 0.1636 | 0.1578 | 0.1636 | 0.1463 | 0.1638 | 0.0189 | 0.0000 | 0.1723 | **0.2357** | 36.81% |
| SeqMatch@100 | 0.2833 | 0.2289 | 0.2395 | 0.2216 | 0.2578 | 0.0209 | 0.0010 | 0.2314 | **0.3268** | 26.76% |
| PPL ↓ | 39.29 | 28.26 | 28.20 | 30.65 | 30.5495 | 149.9894 | > 500.0 | 29.28 | **27.28** | 3.26% |

Table 9: Overall Performance Comparison on LastFM (len=10).

| Metric | FPMC | SASRec | SASRec-ar | STOSA | DiffuRec | DreamRec | PreferDiff | DCRec | **LPDO** | Improve. |
|---|---|---|---|---|---|---|---|---|---|---|
| MeanHR@5 | 0.1093 | 0.1170 | 0.1151 | 0.0894 | 0.1222 | 0.0297 | 0.0035 | 0.1401 | **0.1805** | 28.84% |
| MeanHR@10 | 0.1717 | 0.1611 | 0.1658 | 0.1499 | 0.1734 | 0.0413 | 0.0086 | 0.2099 | **0.2470** | 17.68% |
| MeanHR@20 | 0.2680 | 0.2353 | 0.2347 | 0.2153 | 0.2468 | 0.0623 | 0.0212 | 0.2601 | **0.3200** | 23.03% |
| MeanNDCG@5 | 0.0608 | 0.0615 | 0.0598 | 0.0475 | 0.0633 | 0.0144 | 0.0022 | 0.0721 | **0.0931** | 29.13% |
| MeanNDCG@10 | 0.0694 | 0.0613 | 0.0647 | 0.0630 | 0.0674 | 0.0159 | 0.0041 | 0.0773 | **0.0925** | 19.66% |
| MeanNDCG@20 | 0.0838 | 0.0707 | 0.0693 | 0.0657 | 0.0740 | 0.0194 | 0.0096 | 0.0735 | **0.0899** | 7.28% |
| SeqHR@5 | 0.1080 | 0.1163 | 0.1148 | 0.0888 | 0.1209 | 0.0290 | 0.0000 | 0.1311 | **0.1705** | 30.05% |
| SeqHR@10 | 0.1703 | 0.1606 | 0.1654 | 0.1495 | 0.1715 | 0.0407 | 0.0074 | 0.1881 | **0.2362** | 25.57% |
| SeqHR@20 | 0.2670 | 0.2348 | 0.2341 | 0.2147 | 0.2450 | 0.0606 | 0.0197 | 0.2481 | **0.3113** | 25.47% |
| SeqNDCG@5 | 0.0598 | 0.0608 | 0.0597 | 0.0471 | 0.0625 | 0.0141 | 0.0000 | 0.0673 | **0.0879** | 30.61% |
| SeqNDCG@10 | 0.0686 | 0.0610 | 0.0645 | 0.0624 | 0.0665 | 0.0159 | 0.0033 | 0.0730 | **0.0889** | 21.78% |
| SeqNDCG@20 | 0.0835 | 0.0705 | 0.0690 | 0.0652 | 0.0734 | 0.0187 | 0.0086 | 0.0706 | **0.0880** | 24.65% |
| SeqMatch@50 | 0.0713 | 0.0635 | 0.0566 | 0.0449 | 0.0586 | 0.0098 | 0.0000 | 0.0557 | **0.0762** | 6.87% |
| SeqMatch@100 | 0.1067 | 0.1106 | 0.0957 | 0.0850 | 0.1045 | 0.0107 | 0.0000 | 0.0771 | **0.1162** | 5.06% |
| PPL ↓ | 83.16 | 61.69 | 61.56 | 66.41 | 65.29 | 283.50 | > 500.0 | 61.32 | **60.93** | 0.64% |

Table 10: Overall Performance Comparison on ML-1M (len=5).

| Metric | FPMC | SASRec | SASRec-ar | STOSA | DiffuRec | DreamRec | PerferDiff | DCRec | LPDO | Improve. |
|---|---|---|---|---|---|---|---|---|---|---|
| MeanHR@5 | 0.0158 | 0.0807 | 0.0798 | 0.0865 | 0.0809 | 0.0029 | 0.0094 | 0.1129 | **0.1241** | 9.92% |
| MeanHR@10 | 0.0372 | 0.1372 | 0.1373 | 0.1478 | 0.1548 | 0.0055 | 0.0165 | 0.1887 | **0.2017** | 6.89% |
| MeanHR@20 | 0.0802 | 0.2238 | 0.2249 | 0.2375 | 0.2490 | 0.0099 | 0.0297 | 0.2899 | **0.3100** | 6.93% |
| MeanNDCG@5 | 0.0100 | 0.0455 | 0.0447 | 0.0494 | 0.0510 | 0.0017 | 0.0049 | 0.0632 | **0.0692** | 9.49% |
| MeanNDCG@10 | 0.0181 | 0.0577 | 0.0578 | 0.0629 | 0.0653 | 0.0024 | 0.0073 | 0.0772 | **0.0838** | 8.55% |
| MeanNDCG@20 | 0.0290 | 0.0713 | 0.0722 | 0.0758 | 0.0797 | 0.0034 | 0.0104 | 0.0909 | **0.0963** | 5.94% |
| SeqHR@5 | 0.0157 | 0.0895 | 0.0795 | 0.0865 | 0.0894 | 0.0028 | 0.0004 | 0.1107 | **0.1218** | 10.03% |
| SeqHR@10 | 0.0371 | 0.1365 | 0.1366 | 0.1476 | 0.1540 | 0.0054 | 0.0165 | 0.1823 | **0.1983** | 8.78% |
| SeqHR@20 | 0.0802 | 0.2222 | 0.2232 | 0.2370 | 0.2476 | 0.0098 | 0.0296 | 0.2853 | **0.3051** | 6.94% |
| SeqNDCG@5 | 0.0098 | 0.0434 | 0.0445 | 0.0493 | 0.0506 | 0.0016 | 0.0049 | 0.0621 | **0.0679** | 9.34% |
| SeqNDCG@10 | 0.0179 | 0.0573 | 0.0574 | 0.0627 | 0.0649 | 0.0023 | 0.0073 | 0.0759 | **0.0825** | 8.70% |
| SeqNDCG@20 | 0.0290 | 0.0708 | 0.0716 | 0.0757 | 0.0792 | 0.0034 | 0.0104 | 0.0897 | **0.0949** | 5.80% |
| SeqMatch@50 | 0.0160 | 0.1045 | 0.1049 | 0.1074 | 0.1168 | 0.0000 | 0.0000 | 0.1458 | **0.1559** | 6.92% |
| SeqMatch@100 | 0.0597 | 0.2099 | 0.2017 | 0.2086 | 0.2271 | 0.0000 | 0.0002 | 0.2684 | **0.2796** | 4.17% |
| PPL↓ | 38.43 | 33.62 | 33.58 | 33.47 | 32.47 | 165.59 | > 500.0 | 31.42 | **30.36** | 3.37% |

Table 11: Overall Performance Comparison on ML-1M (len=10).

| Metric | FPMC | SASRec | SASRec-ar | STOSA | DiffuRec | DreamRec | PreferDiff | DCRec | LPDO | Improve. |
|---|---|---|---|---|---|---|---|---|---|---|
| MeanHR@5 | 0.0060 | 0.0584 | 0.0565 | 0.0477 | 0.0632 | 0.0023 | 0.0080 | 0.0757 | **0.0861** | 13.74% |
| MeanHR@10 | 0.0144 | 0.1043 | 0.0996 | 0.0868 | 0.1091 | 0.0043 | 0.0104 | 0.1340 | **0.1482** | 10.60% |
| MeanHR@20 | 0.0368 | 0.1723 | 0.1686 | 0.1473 | 0.1808 | 0.0081 | 0.0208 | 0.2253 | **0.2398** | 6.43% |
| MeanNDCG@5 | 0.0038 | 0.0333 | 0.0323 | 0.0276 | 0.0361 | 0.0013 | 0.0045 | 0.0434 | **0.0486** | 11.98% |
| MeanNDCG@10 | 0.0070 | 0.0451 | 0.0431 | 0.0379 | 0.0464 | 0.0019 | 0.0107 | 0.0575 | **0.0628** | 9.22% |
| MeanNDCG@20 | 0.0141 | 0.0556 | 0.0551 | 0.0483 | 0.0586 | 0.0027 | 0.0101 | 0.0731 | **0.0766** | 4.79% |
| SeqHR@5 | 0.0058 | 0.0581 | 0.0560 | 0.0476 | 0.0624 | 0.0022 | 0.0077 | 0.0717 | **0.0819** | 14.22% |
| SeqHR@10 | 0.0140 | 0.1033 | 0.0985 | 0.0864 | 0.1075 | 0.0043 | 0.0101 | 0.1277 | **0.1419** | 11.12% |
| SeqHR@20 | 0.0361 | 0.1706 | 0.1670 | 0.1467 | 0.1782 | 0.0080 | 0.0206 | 0.2170 | **0.2306** | 6.27% |
| SeqNDCG@5 | 0.0036 | 0.0331 | 0.0320 | 0.0276 | 0.0356 | 0.0012 | 0.0043 | 0.0412 | **0.0461** | 11.89% |
| SeqNDCG@10 | 0.0068 | 0.0446 | 0.0426 | 0.0377 | 0.0456 | 0.0018 | 0.0006 | 0.0550 | **0.0603** | 9.63% |
| SeqNDCG@20 | 0.0139 | 0.0551 | 0.0546 | 0.0480 | 0.0578 | 0.0027 | 0.0101 | 0.0707 | **0.0738** | 4.38% |
| SeqMatch@50 | 0.0000 | 0.0299 | 0.0261 | 0.0214 | 0.0311 | 0.0000 | 0.0000 | 0.0381 | **0.0447** | 17.32% |
| SeqMatch@100 | 0.0032 | 0.0801 | 0.0770 | 0.0614 | 0.0856 | 0.0000 | 0.0000 | 0.0977 | **0.1084** | 10.07% |
| PPL↓ | 85.55 | 68.99 | 70.41 | 73.92 | 68.35 | 305.19 | > 500.0 | 68.58 | **67.44** | 1.66% |

