# OpenReview forum: "Listwise Preference Diffusion Optimization for User Behavior Trajectories Prediction"
_NeurIPS.cc/2025/Conference — NeurIPS 2025 poster_

### Official Review · Reviewer_AK4H · 2025-07-02

**Clarity:** 2
**Significance:** 3
**Originality:** 3
**Rating:** 4
**Confidence:** 4

**Summary:**

This paper focuses on the diffusion-based multi-step recommendation. The authors first define a User Behavior Trajectory Prediction (UBTP) task, which aims to predict the next $k$ items based on the user's historical interactions and the predicted $K$ candidate items at each timestep. The authors then propose a novel Listwise Preference Diffusion Optimization (LPDO) method, which consists of three components: (1) the denoising loss, (2) the regularization (prior matching) loss, and (3) the ListPref loss. Experiments on three datasets validate the effectiveness of LPDO for the UBTP task.

**Questions:**

**Major Concerns:** I have some major concerns as follows:

- **UBTP Task.** This paper introduces a new UBTP task, i.e., $K$-candidate recommendation at the next $k$ steps. While this task is considered novel, the motivation of introducing multi-step recommendation is not well justified. The authors argue that the next-item prediction tends to be short-sighted. However, the autoregressive nature of next-item prediction makes it scalable to multi-step recommendation.
- **LPDO Method.** The proposed LPDO method is based on the Plackett-Luce model (7). However, I have some concerns: (i) Why not simply use the Top-1 Luce's Choice model? There is only one positive item at each step, and the signal from the 2 to $K$-th items is considered useless for the final objective (20). (ii) It seems that the LPDO method is dependent on $K$, i.e., the number of candidate items at each step. For different metrics, e.g., SeqNDCG@$K$, the $K$ should also vary accordingly. However, there is no description of how to set $K$ in the ListPref loss (11), nor any corresponding experiments.
- **Next-item Prediction.** From my understanding, the next-item prediction is essentially a special case of UBTP with $k = 1$. Since this task is more widely studied, the authors should also report the performance of next-item prediction in the experiments.
- **Baselines.** As this paper focuses on the UBTP task, the authors should also provide stronger baselines specifically designed for UBTP. For instance, SASRec can be easily extended to UBTP by incorporating the Plackett-Luce model instead of the Top-1 softmax. The multi-step recommendation can also be achieved in SASRec by changing the embedding size from $d$ to $k \times d$ and predicting the next $k$ items at each step. These baselines should be compared against the proposed LPDO method.
- **Discussion on Existing Methods.** The authors do not provide a comprehensive comparison with existing methods. I recommend the authors to compare LPDO with existing autoregressive recommenders (e.g., SASRec) and diffusion-based recommenders (e.g., PreferDiff, DCRec) on the UBTP task (with $k = 1$ and $k > 1$, respectively).

**Minor Comments:** There are some minor issues/questions that do not affect the overall quality of the paper, including:

- Line 237: There are only three datasets in Table 1, but the text mentions four datasets.
- Line 466: Some typos of $\mathcal{H}$ in Appendix D.3.

**Ethical Concerns:**

["NO or VERY MINOR ethics concerns only"]

**Final Justification:**

The authors' response has addressed my concerns. However, there are still many unclear statements in the paper, such as the inconsistent use of notation, the lack of clarity in the experimental settings, and insufficient explanation of the baseline methods (especially the modified ones). Although I have given a Borderline Accept, I believe that the paper still requires substantial revision to meet the standards for publication.

**Limitations:**

Yes.

**Paper Formatting Concerns:**

N/A.

**Quality:**

2

**Strengths And Weaknesses:**

**Strengths:**

- The authors propose a novel UBTP task that differs from the traditional next-item prediction task.
- The authors introduce several new metrics, including SeqMatch@$K$, SeqHR@$K$, and SeqNDCG@$K$, to evaluate the performance of UBTP.
- The proposed LPDO method is well-designed with rigorous theoretical derivation, achieving superior empirical performance.

**Weaknesses:**

- The motivation of the new UBTP task is not well justified. In other words, why the multi-step is better than the next-item prediction task?
- The necessity of introducing the Plackett-Luce model is also not well justified. Why not simply use the Top-1 Luce's Choice model instead?
- The performance of next-item prediction is not reported, while it is essentially a special case of UBTP with $k = 1$.
- The baselines are not specifically designed for UBTP, and the comparison with existing methods is limited.
- Lack of comprehensive comparison and discussion with existing methods, e.g., SASRec and DCRec.

---

> ### Author Rebuttal · Authors · 2025-07-31
>
> Thank you for your time and effort in reviewing our paper! We sincerely appreciate your thoughtful feedback and hope that our responses effectively address your concerns and support a score revision.
>
> > W1&Q1: The motivation of the new UBTP task is not well justified. In other words, why is the multi-step is better than the next-item prediction task?
>
> Thank you for the question regarding the motivation behind introducing UBTP. While autoregressive next-item prediction can technically be extended to multi-step forecasting by rolling out one prediction at a time, we argue that this greedy, locally optimized generation strategy fails to capture global coherence and long-term dependencies inherent in user trajectories. We try to analyze it from four distinct perspectives:
>
> - **Cascading Error and Exposure Bias:** Autoregressive rollout conditions each future prediction on previously generated items. Errors early in the sequence can propagate, degrading the quality of later predictions [1]. This is particularly problematic in recommendations, where early inaccurate suggestions can distort the entire predicted interaction path.
>
> - **Lack of Global Preference Modeling:** Next-item models optimize local pointwise or pairwise objectives. In contrast, user trajectories are structured objects, where preferences emerge not only at the item level but also from inter-item dependencies, topic continuity, or progression (e.g., watching a trilogy or completing a course). These are not captured by locally greedy methods.
>
> - **Empirical Evidence:** As shown in Tab.1, even strong autoregressive models like SASRec-ar underperform on long-sequence prediction metrics (e.g., SeqMatch), validating that non-autoregressive, trajectory-level modeling yields better alignment with real-world behavior.
>
> - **Non-Autoregressive Benefits:** Our formulation enables parallel generation of full-length trajectories with listwise supervision, offering better sample diversity and efficiency compared to autoregressive alternatives.
>
> Therefore, UBTP is not just a repackaging of next-item prediction, but a principled extension that acknowledges and addresses the limitations of autoregressive rollout when the goal is to model user behavior as a coherent, causally evolving sequence.
>
> ---
>
> > W2&Q2(i):  Why not simply use the Top-1 Luce's Choice model?
>
> Thank you for your thoughtful questions. The introduction of the Plackett–Luce (PL) model is a key design choice motivated by the need to capture global preference structures over full-length trajectories. While simpler alternatives like Top-1 Luce’s Choice model local preference signals, they fail to supervise the relative ordering among multiple relevant items.
>
> Specifically, the PL model allows us to optimize the likelihood of the entire ordered item sequence simultaneously rather than focusing on the top-ranked item at each prediction step. This is essential because in our UBTP task, user preference across future steps exhibits significant ordering constraints and interdependencies. In contrast, the Top-1 model ignores the dependencies among subsequent positions beyond the top-ranked item [2-4], reducing its ability to produce coherent long-term user trajectories. Furthermore, the suggested Top-1 Luce's Choice model essentially corresponds to our baseline method, SASRec-AR [5], in the experiments, where predictions are optimized individually and independently at each step of the sequence.
>
> Our experiments consistently show significant performance gains from the PL loss compared to the position-wise baseline methods (including SASRec-AR). LPDO achieves substantial improvements in all evaluation metrics, demonstrating that explicitly modeling the full-sequence ranking structure is necessary and beneficial for accurate multi-step user behavior prediction.
>
> ---
>
> > W2&Q2(ii): It seems that the LPDO method is dependent on K, i.e., the number of candidate items at each step. For different metrics, e.g., SeqNDCG@K, the K should also vary accordingly.
>
> We would like to clarify that LPDO is not dependent on the number of candidate items at each step. During the optimization process, LPDO interacts with all candidate items for optimization without negative sampling. Specifically, the ListPref loss itself is formulated to consider the full candidate set, and its computation is independent of the evaluation metric.
>
> Regarding evaluation metrics, following traditional sequential recommendation settings [5-7], the value of K in metrics such as SeqNDCG@K is determined by the evaluation protocol and does not affect the training procedure or the ListPref loss (Eq.11). In our experiments, we follow standard practice by reporting results for multiple values of K (e.g., K=5, 10, 20), and these are clearly specified in the experimental section.
>
> ---
>
> > W3&Q3: From my understanding, the next-item prediction is essentially a special case of UBTP with k=1. Since this task is more widely studied, the authors should also report the performance of next-item prediction in the experiments.
>
> Thanks for your insightful feedback. Indeed, next-item prediction is a special case of UBTP when the prediction length k = 1, and it corresponds to the widely studied task of Sequential Recommendation (SR). UBTP requires the model to generate a coherent sequence of future interactions, capturing inter-item dependencies and preference evolution over multiple steps, which are not addressed in standard next-item recommendation settings.
>
> To address the reviewer’s concern, we report the performance on the first position of the predicted trajectory (i.e., the first item among the multi-step prediction) using models trained under the UBTP setting. This provides a reference point that overlaps with next-item prediction. We report the performance on the first position (i.e., the next item) from the predicted trajectory. The results on MovieLens-1M (len=5), evaluated by HR@5 and NDCG@5, are shown below:
>
> |Metric|SASRec|STOSA|DiffuRec| DCRec|LPDO|
> |-|-|-|-|-|-|
> | HR@5|0.0921|0.0912|0.1052|0.1293|0.1571|
> | NDCG@5|0.0518|0.0506|0.0621|0.0710|0.0866|
>
> LPDO achieves the highest performance on the first predicted item, suggesting that even when evaluated on short-horizon prediction, the model trained for UBTP generalizes well. We will include these results in the appendix of the revised version.
>
> ---
>
> > W4&Q4: As this paper focuses on the UBTP task, the authors should also provide stronger baselines specifically designed for UBTP.
>
> Thank you for your insightful suggestions. We would like to clarify that in our experiments, we have already extended SASRec to support UBTP by applying the Top-1 softmax at each prediction step (as SASRec-ar), enabling multi-step recommendation. This approach aligns with the UBTP setting and provides a fair baseline for comparison.
> Besides, we include sequential recommendation baselines such as DiffuRec and DreamRec, which have been widely used and offer valuable points of comparison. These baselines help demonstrate the effectiveness of our method.
>
> Regarding the embedding size, our implementation does not require changing the embedding dimension from d to d×k. Instead, the search space expands from d to $d^k$, where d is the number of candidate items and k is the prediction length. This design ensures that the model can generate listwise predictions without modifying the underlying embedding structure.
>
> Regarding incorporating the Plackett-Luce optimization ($\gamma=0.5$) with SASRec, we report the performance comparison below:
>
> | Model|SHR@5↑|SNDCG@5↑|SHR@10↑|SNDCG@10↑|SMatch@50↑|SMatch@100↑|PPL↓|
> |-|-|-|-|-|-|-|-|
> |SASRec|0.0805|0.0453|0.1364|0.0573|0.1045|0.2009|33.62|
> |SASRec+PL (γ=0.0)|0.0824|0.0486|0.1446|0.0617|0.1143|0.2265|32.83|
> |SASRec+PL (γ=0.3)|0.0852|0.0480|0.1469|0.0624|0.1139|0.2230|32.60|
> |SASRec+PL (γ=0.5)|0.0867|0.0487|0.1473|0.0626|0.1152|0.2286|32.12|
> |SASRec+PL (γ=0.8)|0.0848|0.0473|0.1468|0.0620|0.1128|0.2192|32.12|
> |LPDO|0.1218|0.0679|0.1983|0.0825|0.1559|0.2796|30.36|
>
> This experiment shows that incorporating PL optimization brings performance improvements to existing models such as SASRec. However, LPDO still outperforms these variants by a clear margin across all metrics, demonstrating that our method benefits not only from listwise supervision, but also from its integration into the diffusion-based trajectory modeling framework.
>
> ---
>
> > W5&Q5: Discussion on Existing Methods. The authors do not provide a comprehensive comparison with existing methods. I recommend the authors to compare LPDO with existing autoregressive recommenders (e.g., SASRec) and diffusion-based recommenders on the UBTP task (with K=1 and K > 1).
>
> Thank you for raising this point. In our experiments, we include SASRec with autoregressive mode (denoted as SASRec-ar), as well as diffusion-based baselines such as DiffuRec and DCRec. We provide the comparion results of K=1 in the response of W3&Q3, please refer to it. As for results of K=3/5/10, please refer to Tab.1.
>
> We appreciate your feedback. If there are additional aspects you would like us to elaborate on, we are happy to provide further analysis.
>
> ---
>
> > Q6: Typo at Line 237 and 466
>
> Thank you for pointing out. We will revise the manuscript to correct this.
>
>
> [1] Limitations of Autoregressive Models and Their Alternatives
>
> [2] Fast and accurate inference of Plackett–Luce models
>
> [3] Plackett-Luce model for the learning-to-rank task
>
> [4] RankDistil: Knowledge Distillation for Ranking
>
> [5] Self-Attentive Sequential Recommendation
>
> [6] DiffuRec: A Diffusion Model for Sequential Recommendation
>
> [7] Generate What You Prefer: Reshaping Sequential Recommendation via Guided Diffusion

---

> ### Comment · Reviewer_AK4H · 2025-08-01
> **Response to Authors**
>
> Thanks for the authors' detailed rebuttal. While the rebuttal partially resolved my concerns, I still have some questions.
>
> The definition of ListPref is quite confusing. The ListPref loss (11), or the Soft-ListMLE loss (9), needs the full ground-truth ranking label for the $K$ candidates $A _ {u, j}$ at time step $j$. However, in sequential recommendation, there is only one positive item $i _ {u, n + j}$ at each time step $j$. Therefore, the full ground-truth ranking label can only be one-hot, i.e., the positive item $i _ {u, n + j}$ should be ranked at the top, and the others can be ranked arbitrarily, no matter how many candidates $K$ are predicted. From this perspective, considering there is no need to capture the "relative ordering among multiple relevant items" (since there is only one relevant item), I question the authors' response to Q2(i).
>
> In Q2(ii), the authors mentioned that "LPDO interacts with all candidate items for optimization without negative sampling". While I agree with this statement, I still concern that why ListPref (11) is independent of the number of candidates $K$ that predicted at each step? It is apparent that $K$ appears in the Eq. (11).
>
> I may reconsider my rating if the authors can clarify the above issues, and I thank the authors again for their efforts in rebuttal.

---

> ### Author Response · Authors · 2025-08-04
> **Response to Reviewer AK4H (Part 1)**
>
> Thank you for your valuable comments and thoughtful follow-up. We appreciate the opportunity to clarify the formulation of ListPref in the context of our task.
>
> > The definition of ListPref is quite confusing. The ListPref loss (11), or the Soft-ListMLE loss (9), needs the full ground-truth ranking label for the $K$ candidates $A_{u,j}$ at time step $j$. However, in sequential recommendation, there is only one positive item $i_{u,n+j}$ at each time step $j$. Therefore, the full ground-truth ranking label can only be one-hot, i.e., the positive item should be ranked at the top, and the others can be ranked arbitrarily, no matter how many candidates $K$ are predicted. From this perspective, considering there is no need to capture the ‘relative ordering among multiple relevant items’ (since there is only one relevant item), I question the authors’ response to Q2(i).
>
> We understand the reviewer’s concern, and we believe it stems from a possible mismatch between two task formulations. We greatly appreciate the chance to further distinguish our setting from the conventional next-item recommendation task.
>
> 1. Step-by-step next-item recommendation
>    - Definition: At each time step $j$, predict only the single next item $i_{u,n+j}$.
>    - Ground-truth label: One-hot (one positive, all others negative).
>    - Implication: Listwise losses over a one-item list reduce to a Top-1 choice model.
>
> 2. Whole-sequence prediction (our setting)
>    - Definition: Predict the entire length-$K$ future trajectory $\bigl(i_{u,n+1},i_{u,n+2},\dots,i_{u,n+K}\bigr)$ as one ordered list.
>    - Ground-truth label: A full permutation of $K$ distinct positive items—not a single one-hot.
>    - Implication: ListPref (Eq.(11)) directly maximizes the Plackett–Luce likelihood of this complete sequence, requiring and utilizing a full ranking of $K$ positives.
>
> **Why the one-hot critique does not apply in our setting for UBTP task.**
>
> - Full-sequence ground truth: In our UBTP task, the target is the ordered sequence $\bigl(i_{u,n+1},\dots, i_{u,n+K}\bigr)$. We therefore have exactly one positive per rank, for a total of $K$ positives, which constitutes a full ranking label.
> - ListPref’s mechanism: ListPref still performs sequential softmax draws over all $K$ candidates, where each position’s softmax is conditioned on all previous positions. This setting enable rich comparisons at each draw (positive vs. negatives and negative vs. negative), but now over the complete sequence rather than a single next-item step.
> - No one-hot reduction: Unlike stepwise training, our formulation never reduces to “one positive vs. all negatives” in isolation; it always conditions on previously drawn items, fully leveraging the permutation.
>
> **Empirical validation.**
>
> In Sec.5.2 (Tab.1), LPDO’s ListPref loss outperforms the position-wise CE baseline (SASRec-ar). To further validate this, we provide a performance comparison on the first position of the prediction sequence on MoiveLens-1M (target len=5) as below. Our LPDO also surpasses SASRec-ar in this setting.  These gains confirm that, under whole-sequence prediction, ListPref harnesses the full permutation of $K$ positives to deliver superior ranking quality.
>
> |Metric|SASRec-ar|LPDO|
> |-|-|-|
> | HR@5|0.0954|0.1571|
> | NDCG@5|0.0558|0.0866|
>
> In addition to empirically validating that our proposed loss achieves higher HR@5 and NDCG@5 compared to position-wise CE loss, we would also like to highlight our contributions beyond loss formulation. While the ListPref loss is a component of our design, the key contribution of LPDO also includes integrating listwise supervision within a non-autoregressive diffusion framework for multi-step trajectory generation. This enables:
> - Coherent prediction across multiple future steps.
> - Jointly optimized reconstruction + ranking via a principled ELBO.
> - Causal modeling via transformer backbones.
> To our knowledge, this is the first work to bridge listwise learning and diffusion-based trajectory prediction, moving beyond conventional autoregressive or pointwise paradigms.

---

> ### Author Response · Authors · 2025-08-04
> **Response to Reviewer AK4H (Part 2)**
>
> > In Q2(ii), the authors mentioned that "LPDO interacts with all candidate items for optimization without negative sampling". While I agree with this statement, I still concern that why ListPref (11) is independent of the number of candidates predicted at each step? It is apparent that K appears in the Eq. (11).
>
> Thank you for the suggestion related to our presentation of the $K$.
>
> In the standard sequential-recommendation setting (e.g., DiffuRec[1], DreamRec[2]), they adopt all items in the dataset for both training and evaluation. We follow this setting in our paper, and the notation $K$ refers to using the entire item set (i.e., $K$ = all items) as candidates for optimizing the predicted items.
>
> However, this $K$ is different from the @K used in evaluation metrics (e.g., SNDCG@K, SeqMatch@K). In these metrics, @K denotes the number of top-ranked items considered when evaluating whether the ground-truth item (i.e., the item the user actually interacted with) appears in the recommended list. In the UBTP task, each predicted results in the sequence has its own top@K items.
>
> We sincerely appreciate the reviewer for pointing this out, and we will revise the notation from @K to @N in the updated manuscript to avoid confusion and improve clarity.
>
> [1] DiffuRec: A Diffusion Model for Sequential Recommendation
>
> [2] Generate What You Prefer: Reshaping Sequential Recommendation via Guided Diffusion

---

> > ### Comment · Reviewer_AK4H · 2025-08-04
> > **Response to Authors' Additional Rebuttal**
> >
> > I appreciate the authors' additional response, which addressed my concerns, and I have increased my score accordingly.

---

### Official Review · Reviewer_EHty · 2025-07-02

**Clarity:** 2
**Significance:** 2
**Originality:** 3
**Rating:** 4
**Confidence:** 4

**Summary:**

This article proposes a novel diffusion based framework LPDO (List Preference Diffusion Optimization) for user behavior trajectory prediction, which uses a generative paradigm to predict users' future interaction trajectories. The author believes that existing diffusion based recommendation models largely handle each position independently, thereby ignoring global list dependencies. The LPDO proposed in this article addresses this limitation by directly incorporating Plackett Luce list based ranking targets into diffusion ELBO, thereby aligning the generation process with trajectory level preferences. It also introduced a new metric, SeqMatch, for rigorous trajectory level evaluation and conducted extensive experiments, achieving the most significant results compared to strong baselines on four datasets.

**Questions:**

Please refer to Weaknesses section.

**Ethical Concerns:**

["NO or VERY MINOR ethics concerns only"]

**Limitations:**

Yes

**Paper Formatting Concerns:**

None observed

**Quality:**

3

**Strengths And Weaknesses:**

Strengths：
1. The ELBO formulation incorporating Plackett-Luce likelihood provides a principled bridge between generative denoising and listwise ranking. The derivation of the tight variational bound is a notable theoretical contribution.
2. The List Preference Diffusion Optimization (LPDO) framework proposed in this article is innovative in integrating list preference information into diffusion models. This method aligns the backpropagation process with the list ranking target, which is a new approach to improve the consistency and accuracy of predicting user behavior trajectories.
3. LPDO achieves ​faster convergence and enables ​single-step inference​, reducing inference latency.

Weaknesses:
1. From Figure 3 (b), it is shown that the prediction results for long-term goals are similar to those for short-term goals, and LPDO shows a similar decrease in this prediction result compared to other sequence recommendation methods (pos1-pos5). This does not seem to reflect the superiority of LPDO in modeling users' long-term preferences compared to other methods.
2. The expression in the paper is not clear enough, such as the "UBTM" in line 149, which is not explained in the entire text.
3. The experimental results of various models at different prediction lengths are shown in Table 1. LPDO did not show a significant increase in advantages compared to other models at longer lengths (len=10) and shorter lengths (len=5)
4. ​Hyperparameter Sensitivity: Performance heavily depends on manual tuning of loss weight λ and penalty γ. Optimal γ varies across datasets (0.0–0.8), raising generalizability questions. An adaptive tuning strategy or ablation on sensitivity is missing.
5. The model requires ​fixed-length trajectories during training​. Real-world sequences are variable-length, limiting applicability to open-ended scenarios.
2. Why is it that setting the diffusion step size to 50 and the sampling step size to 1 can yield very significant results?

---

> ### Author Rebuttal · Authors · 2025-07-31
>
> We greatly appreciate your time and effort in reviewing our work. Your comments were very helpful, and we have provided detailed responses below to clarify and address the issues you raised.
>
> > W1: From Figure 3 (b), it is shown that the prediction results for long-term goals are similar to those for short-term goals, and LPDO shows a similar decrease in this prediction result compared to other sequence recommendation methods (pos1-pos5). This does not seem to reflect the superiority of LPDO in modeling users' long-term preferences compared to other methods.
>
> Thank you for the insightful comments. We try to address these concerns from two distinct perspectives:
>
> - **Performance decay is inherent in the UBTP task.**  Compared to one-step prediction, UBTP is intrinsically more challenging due to error accumulation over multiple steps. As the model predicts further into the future, uncertainty compounds, and performance naturally declines due to error propagation. This is a well-known phenomenon in multi-step sequence modeling, such as the NLP task [1]. Despite this, LPDO demonstrates greater robustness and slower degradation than other methods, underscoring its effectiveness.
>
> - **A similar performance trend does not equate to having similar modeling capabilities.** While it is true that all models, including LPDO, demonstrate a decreasing accuracy trend across future positions, this observation alone does not imply comparable modeling capacity. As illustrated in Fig.3(b), LPDO consistently outperforms all baselines at each prediction step, including the later stages. This consistent advantage highlights LPDO’s superior ability to capture both short-term signals and long-term user preferences.
>
> We thank the reviewer for the insightful comment and acknowledge that there is still potential for enhancing the UBTP task in future work.
>
> ---
>
> > W2: The expression in the paper is not clear enough, such as the "UBTM" in line 149.
>
> Thank you for pointing this out. ‘UBTM’ is a typo and should be ‘UBTP’. We will revise the manuscript to correct this.
>
> ---
>
> > W3: LPDO did not show a significant increase in advantages compared to other models at longer lengths (len=10) and shorter lengths (len=5).
>
> Thank you for bringing this point to us. We would like to clarify that LPDO consistently demonstrates substantial performance improvements across both shorter (len=5) and longer (len=10) future sequence prediction as shown in Tab.1.
>
> For instance, on the Last-FM (target len=5) setting, LPDO achieves a 29.83% improvement over the best baseline in SeqHR@5, and a 36.81% absolute gain in SeqMatch@50, which measures strict trajectory-level accuracy. On the LastFM (target len=10) dataset, LPDO improves SeqHR@5 by 30.05% and SeqMatch@50 by 6.87\% over the best baseline. These gains remain substantial and statistically significant, and they underscore LPDO’s robustness across different settings.
>
> ---
>
> > W4: Performance heavily depends on manual tuning of loss weight $\lambda$ and penalty $\gamma$.
>
> Thank you for bringing this point to us. We would like to clarify that loss weight $\lambda$ can be set to 0.1 for all experiment settings to achieve superior performance, as mentioned in Sec.5.1. As for the penalty factor $\gamma$, as shown in Fig.3(c) and the additional results below, all different $\gamma$ settings consistently outperform the baselines in Tab.1, demonstrating the robustness of our method to this hyperparameter.
>
> **Table: MovieLens-1M (len=5)**
> | Metric      | SASRec | DiffuRec | DCRec  | LPDO($\gamma$=0.0) | LPDO($\gamma$=0.3) | LPDO($\gamma$=0.5) | LPDO(γ=0.8) |
> |-------------|--------|----------|--------|-------------|-------------|-------------|-------------|
> | SHR@5       | 0.0895 | 0.0894   | 0.1107 | 0.1190      | 0.1218      | 0.1184      | 0.1164      |
> | SNDCG@5     | 0.0434 | 0.0506   | 0.0621 | 0.0637      | 0.0679      | 0.0651      | 0.0641      |
> | SMatch@50   | 0.1045 | 0.1074   | 0.1458 | 0.1509      | 0.1559      | 0.1558      | 0.1523      |
>
> **Table: LastFM (len=5)**
> | Metric      | SASRec | DiffuRec | DCRec  | LPDO($\gamma$=0.0) | LPDO($\gamma$=0.3) | LPDO($\gamma$=0.5) | LPDO($\gamma$=0.8) |
> |-------------|--------|----------|--------|-------------|-------------|-------------|-------------|
> | SHR@5       | 0.1816 | 0.1793   | 0.1931 | 0.2367      | 0.2388      | 0.2403      | 0.2507      |
> | SNDCG@5     | 0.0917 | 0.0906   | 0.0972 | 0.1150      | 0.1213      | 0.1211      | 0.1260      |
> | SMatch@50   | 0.1578 | 0.1638   | 0.1723 | 0.2205      | 0.2292      | 0.2300      | 0.2357      |
>
>
> We will add additional results in the revised version to improve the clarity of $\gamma$ settings. Thank you for your suggestion regarding adaptive tuning. We will consider incorporating adaptive strategies in future work.
>
> ---
>
> > W5: The model requires fixed-length trajectories during training. Real-world sequences are variable-length, limiting applicability to open-ended scenarios.
>
> Thank you for raising this concern. We would like to clarify that our model can support variable-length sequences during training. Our model is designed for the UBTP task (next-sequence prediction), which is an extension of the next-item prediction setting in sequential recommendation. In this field, it is standard practice to use sequences with a maximum length for training, and variable-length sequences are typically handled by padding shorter sequences to the maximum length, which is a widely adopted and effective strategy in sequence modeling. Therefore, our method can naturally support variable-length trajectories without loss of generality or applicability. During model inference, our model can also support variable-length sequences.
>
> Thus, our method remains applicable to open-ended scenarios and real-world deployment. We will clarify this implementation detail in the revised version.
>
> ---
>
> > W6: Why is it that setting the diffusion step size to 50 and the sampling step size to 1 can yield very significant results?
>
> Thank you for your question. Compared to CV [2] and NLP [3] tasks, recommendation targets lie in a low-dimensional, discrete, and highly structured space (e.g., item indices). The conditional signals, such as user history, are strong and tightly constrain the output manifold. As a result, even coarse denoising with a single sampling step can reliably recover high-quality item representations for ranking. In contrast to NLP/CV tasks requiring fine-grained generation, the discrete and sparse nature of recommender outputs makes them significantly more robust to approximation during fast sampling. Meanwhile, our observation is consistent with prior works such as DiffRec[4], SdifRec[5], and DCRec[6], which also report that a small number of steps can yield competitive results in recommendation tasks. We will clarify this point in the revised version.
>
> [1] Scheduled Sampling for Sequence Prediction with Recurrent Neural Networks
>
> [2] High-Resolution Image Synthesis with Latent Diffusion Models
>
> [3] DiffuSeq: Sequence to Sequence Text Generation with Diffusion Models
>
> [4] Diffusion Recommender Model
>
> [5] Bridging User Dynamics: Transforming Sequential Recommendations with Schrödinger Bridge and Diffusion Models
>
> [6] Dual Conditional Diffusion Models for Sequential Recommendation

---

> > ### Author Response · Authors · 2025-08-06
> > **Looking Forward to Your Further Comments**
> >
> > Thank you again for your valuable feedback on our submission. Your  suggestions are thoughtful and insightful, and we hope our responses adequately address your concerns.
> >
> > As the discussion deadline approaches, we would greatly appreciate any further feedback you may have. If you have additional questions or concerns, please feel free to share them. We remain open and happy to engage in further discussion.

---

### Official Review · Reviewer_ULMe · 2025-07-02

**Clarity:** 4
**Significance:** 3
**Originality:** 3
**Rating:** 5
**Confidence:** 4

**Summary:**

This work tackles the complex problem of forecasting user behaviour trajectories multiple steps into the future, a problem distinct from regular sequential recommendation. Generative recommendation with diffusion models is becoming more and more prevalent across the recommender systems research community due to the ability of such models to model uncertainty and learn latent data distributions in order to produce diverse recommendations. However, diffusion models are typically currently limited to next-item recommendation settings due to their inability to account for the dynamic nature of user preferences across stages of a predicted user behaviour trajectory. Point-wise preference signals can be incorporated, but joint dependences and relative ordering of items is still ignored, two factors which considerably impact the quality and impact of a sequence of recommendations. This work aims to address these issues directly.

The first contribution provided by this work is the formalisation of this task (UBTP) of forecasting an ordered sequence of $k$ interactions. In this problem setting, the authors propose a novel approach termed LDPO which integrates Plackett-Luce ranking terms into the evidence lower bound (ELBO) of a non-autoregressive diffusion model in continuous latent space. Deriving this tractable lower bound which jointly handles reconstruction and listwise likelihoods, whilst simultaneously ensuring stable training and managing the generation fidelity-preference alignment trade-off is a non-trivial task, which the authors tackle through devising an end-to-end differentiable scoring head mapping denoised latents to logits and formulating a tight decomposition of the ELBO. Additionally, the authors also introduce a new metric, SeqMatch, which is specifically designed for evaluating multi-step recommendation quality. Specifically, SeqMatch@$k$, computes the percentage of test user trajectories where each item in the trajectory is present in the model’s top $k$ predictions for their respective positions.

Experimental results are provided exploring the performance of LPDO compared to competing baselines and to other diffusion-based methods, with some additional studies performed on the impact of different hyperparameters on performance. Several large-scale, widely used recommendation benchmark datasets are used for evaluation, and a range of both traditional and generative recommendation algorithms are evaluated alongside LPDO. The authors find that their approach outperforms the selected baselines across the board and that LPDO converges faster than other diffusion-based methods.

**Questions:**

**Q1** - As mentioned above, I have very little to ask as the paper is very thorough, and I recommend acceptance. One question I do have is whether the authors could expand further on the rationale behind not providing open-source access to the code for the implementation? The high potential for impact that this paper has could be limited somewhat by not providing a base implementation for further research to easily progress onwards from.

**Q2** - The authors note in the checklist that the improvements in performance are statistically significant, but this is not included in the main text. Could you confirm that this is the case for all experiments conducted, and if not, could these cases be flagged in the results table (e.g. via an asterisk or simply un-bolding the result)?

**Ethical Concerns:**

["NO or VERY MINOR ethics concerns only"]

**Final Justification:**

The authors have addressed my concerns regarding code availability and statistical significance during the rebuttal phase, but my overall assessment of the paper remains unchanged from my original score.

**Limitations:**

Yes

**Quality:**

4

**Strengths And Weaknesses:**

**Strengths**
* Overall, the work is novel and well-motivated. Diffusion models may be gaining prevalence for large-scale recommendation, however tackling this problem of multi-step trajectory forecasting in a manner which properly accounts for the ordering and causal relationships between items in a given trajectory is a key step in ensuring these methods are more widely applicable in real-world scenarios. This work clearly appears to be a considerable step towards addressing this problem, and as such has potential for significant impact.
* The paper itself is very well structured and clear throughout; aspects of the model formulation, derivation of the variational lower bound and the training procedure are all outlined concisely and clearly, and on inspection all the work appears sound and of high quality. Likewise, the evaluation of the model and the competing methods is presented in an easily parsed and engaging manner, with the key takeaways being clearly signposted and explained in depth.
* The empirical results are very good, with the proposed technique in some cases providing very significant improvements over the competing baselines, across all of the datasets used for testing. The evaluation has been thoroughly carried out, with a range of different metrics used to show the superior performance of the proposed method.
* A particular strength of this work is the choice by the authors to formulate not only the task being studied, but also a more relevant metric for evaluation. Often, having a clear target, problem statement and metric for improvement is the most important factor in driving forward progress in a certain area of research, and I think contributions such as this which attempt to provide these things can be highly impactful in the sense that they incentivise future work on the topic, driving further developments.

**Weaknesses**
* Ultimately, I have very few negative things to say about the paper. It’s very well constructed and presented, is novel, addresses a clear problem, and evidences the claims included with thorough evaluation.
* One clear weakness is the lack of open-source code, as this will limit the impact of the paper given that it is more challenging for practitioners to use the technique, and more time consuming for researchers to pursue further work based on this approach. I appreciate that this may not be possible for certain reasons, however it does certainly limit the impact.
* As mentioned in the questions section below, some further clarity about which of the experimental results are statistically significant would be appreciated, as this is not immediately clear from the results tables.

---

> ### Author Rebuttal · Authors · 2025-07-31
>
> We sincerely thank the reviewer for the detailed, thoughtful, and positive evaluation of our work. We are especially grateful for your recognition of our contributions. Your comments on the clarity of presentation, strength of empirical results, and potential impact are highly encouraging and greatly appreciated.
>
> > W2, Q1: The lack of open-source code.
>
> Thank you for raising this point. We plan to release the data and code after the paper is published.
>
> ---
>
> > W3, Q2:  Some further clarity about which of the experimental results are statistically significant would be appreciated.
>
> Thank you for the suggestion. We confirm that the improvements are statistically significant for the main performance results (Sec. 5.1), factor analysis (Sec. 5.2), and ablation study (Sec. 5.3). We will clarify this in the revised manuscript and ensure that statistical significance is reported for all other experiments (in appendix).

---

> > ### Comment · Reviewer_ULMe · 2025-08-04
> > **Response to rebuttal**
> >
> > I appreciate the authors commitment to releasing the data and code, and also for clarification of the statistical significance of the results. My overall assessment of the paper remains unchanged.

---

### Official Review · Reviewer_WfsB · 2025-07-03

**Clarity:** 2
**Significance:** 3
**Originality:** 2
**Rating:** 4
**Confidence:** 3

**Summary:**

* This paper considers recommender systems and has three contributions:
    * It proposes a new problem, User Behavior Trajectory Prediction (UBTP), of predicting a sequence of a user's preferred items in the future k time steps by using this user's history of viewed items in the past n time steps.
    * It adopts a diffusion model to do such prediction, proposing a method called Listwise Preference Diffusion Optimization (LPDO). The diffusion model's output is k embeddings, and each embedding is a vector of utilities for all the items. For each of the k future time steps, the system predicts the user's top-K items according to the output of the diffusion model.
    * It proposes a new metric to evaluate the performance of UBTP, by checking how often the top-K predicted items actually contain the true most-preferred items.
* This paper evaluated LPDO using 4 real-world datasets, where k is set to 3, 5, or 10, against 7 baselines. It shows that LPDO outperformed significantly all of them.

**Questions:**

* Line 53-59 tries to explain why the proposed method LPDO outperforms previous methods, but are too technical to fully grasp in the introduction section. Fig.1 clearly explains the insight of why Fig.1(c) is beneficial, but is not echoed by the technical presentation of LPDO. It could be helpful in the later section of the paper to explain why LPDO's new loss function can achieve Fig.1(c), improving upon previous work in Fig.1(b).
* Line 153 mentioned that the paper follows [18] to also include history H as part of X0, which the model is trying to generate. I am wondering why this is helpful. Could you provide some insights to explain this design consideration?
* Line 157 says that pθ(X_{t−1} | Xt, H). I thought that for each t, the history H is already a part of Xt. So why do we have the extra "H" in `pθ(X_{t−1} | Xt, H)`?
* Eq.11 introduces the loss for listwise preference. However, it seems that `L_ListPref` is defined for a particular future time step j? How would the system combine all the k future time steps in the loss, considering that the system is trying to capture the causal relation between those k future time steps?
* Eq.16 contains both reconstruction loss and list preference loss. I thought the reconstruction loss already contains the list preference loss, as the reconstruction loss is about the loss of the reconstructed k future step predictions?

**Ethical Concerns:**

["NO or VERY MINOR ethics concerns only"]

**Final Justification:**

This paper proposes to use a diffusion model to predict future users' preferred items. The rebuttal clarified my questions. I think it contributes a new tool to user trajectory prediction. I will keep my rating.

**Limitations:**

Yes

**Quality:**

3

**Strengths And Weaknesses:**

* Strength
    * The proposed problem is well-motivated.
    * The proposed metric aligns well with the proposed problem.
    * The proposed algorithm empirically significantly outperformed baselines.
* Weakness
    * One concern is that the key innovation and insights behind the proposed method, LPDO, are a bit unclear, making it hard to fully appreciate its technical novelty. For more details, please refer to the "Questions" section later in the review.
    * Another concern is that the empirical analysis is a bit unclear about the key contributor to the improved performance of the proposed method. I think there could be a couple of factors why the proposed method outperformed previous methods: (1) it tries to predict k future time steps, rather than 1 time step; (2) its novel loss function; (3) it uses the causal transformer. It could be helpful to better understand the contribution of these factors. A better understanding of this question could allow the readers to appreciate the technical novelty more.
        * Also, related to factor (1), since the proposed method predicts a longer future, would it require more data or a longer time to train? Understanding this question could allow potential users to better decide when to use the proposed method.
* Minor
    * Line 119: "target item" is confusing, as it makes me confused whether this refers to the single-item prediction task or the proposed UBTP task, which tries to predict k items.
    * Line 158: `Equation (24)` is mistakenly referred I think.
    * Line 190: `Section 3` is mistakenly referred I think.

---

> ### Author Rebuttal · Authors · 2025-07-31
>
> We are truly grateful for the time and effort you dedicated to reviewing our work. We hope our detailed responses have clarified the issues raised.
>
> > W1&Q1: The key innovation and insights are a bit unclear. Lines 53–59 are too technical for the introduction. Fig.1(c) is not echoed by the technical presentation.
>
> Thank you for your thoughtful feedback and suggestions. The key advantage of our proposed LPDO method is that it directly optimizes the listwise preference of item sequences as shown in Fig.1(d), rather than optimizing individual items independently as in pointwise methods. This listwise optimization allows LPDO to better capture the dependencies and overall ranking quality of the entire sequence, leading to improved recommendation performance. Figure 1(c) is meant to conceptually illustrate how our preference-aware diffusion model focuses the prediction distribution more effectively on semantically related targets. We elaborate this effect formally in Section 4.2 (Bridging Diffusion and Sequential Listwise Ranking) and Section 4.4 (Connecting Diffusion Models With Listwise Maximum Likelihood Estimation), where we introduce the ListPref loss and the ELBO formulation. By aligning the diffusion denoising steps with the Plackett–Luce listwise supervision signal, the model is guided to concentrate generation probability on ranked, preference-consistent item sequences, thereby reducing the dispersion toward unrelated content. We will clarify this in revised version.
>
> ---
>
> > W2:  The empirical analysis is a bit unclear about the key contributor to the improved performance of the proposed method. There could be a couple of factors why the proposed method outperformed previous methods: (1) k-step prediction; (2) novel loss function; (3) causal transformer.
>
> Thank you for raising this point. As you mentioned, the performance improvement of LPDO mainly comes from three factors:
>
> - **Designed for k-step prediction**: LPDO is specifically designed for the k-step prediction task. While all baselines are extended to support prediction over k future time steps for fair comparison, LPDO is the first model explicitly designed for the UBTP task, addressing the limitations of directly applying next-item prediction models in this setting.
>
> - **The novel loss function**: We proposed a novel listwise preference-aware loss. Our loss maximizes the
> joint likelihood of the entire ordered sequence, better capturing the ordering and dependencies among items. We have provided an ablation study of different loss components in LPDO in Tab.2. The results demonstrate the effectiveness of our loss.
> Furthermore, we replace the original list preference loss in LPDO with traditional cross-entropy loss for directly comparison. The results on ML-1M (target length=5) presented below indicate the importance of list-wise optimization in the UBTP task.
>
> |Loss type|SHR@5 ↑|SNDCG@5 ↑|SHR@10 ↑|SNDCG@10 ↑|SMatch@50 ↑|SMatch@100 ↑|PPL ↓ |
> |-|-|-|-|-|-|-|-|
> |Cross Entropy|0.1103|0.0645|0.1914|0.0801|0.1477| 0.2659|31.35|
> |ListPref|0.1218|0.0679|0.1983|0.0825|0.1559|0.2796|30.36 |
>
> - **Applying causal transformer**: Regarding transformer modules, we provide a comparison between different transformer modules, including bidirectional [1], prefix [2], causal [3], on ML-1M (target length=5) as below. The results show that causal transformer outperforms the other two types, which indicates the importance of capturing performance and causality among user iterations.
>
> | Transformer|SHR@5 ↑|SNDCG@5 ↑|SHR@10 ↑|SNDCG@10 ↑|SMatch@50 ↑|SMatch@100 ↑|PPL ↓ |
> |-|--|-|-|-|-|-|-|
> |Bidirection|0.1159|0.0662|0.1925|0.0806|0.1500|0.2709|31.35|
> |Prefix|0.1175|0.6517|0.1927|0.7966|0.1521|0.2740|30.86|
> |Causal|0.1218|0.0679|0.1983|0.0825|0.1559|0.2796|30.36|
>
> ---
>
> > W3:  Since the proposed method predicts a longer future, would it require more data or a longer time to train?
>
> Thank you for bringing up this point. It is worth noting that both LPDO and other baselines demand more training data to reach optimal performance. This is largely because the UBTP task entails a significantly larger solution space than next-item prediction, expanding from $n$ to $n^k$, where $n$ is the number of candidate items and $k$ is the number of steps to predict.
>
> To further investigate this issue, we analyze model performance (measured by SeqMatch@50) on the ML-1M dataset (target length = 5) under varying training set sizes: the full dataset, 3,000 samples, and 1,000 samples. As shown below, model performance improves consistently with more training data. Notably, LPDO consistently outperforms all baselines across all three data settings.
>
> | Num of data | SASRec | DiffuRec | DCRec  | LPDO   |
> |-------------|--------|----------|--------|--------|
> | All         | 0.1045 | 0.1168   | 0.1458 | 0.1559 |
> | 3000        | 0.0973 | 0.1097   | 0.1280 | 0.1334 |
> | 1000        | 0.0548 | 0.0599   | 0.0671 | 0.0849 |
>
> Regarding training time, as shown in Fig. 3(a), our model converges faster than the baseline due to LPDO`s optimization for global sequence-level preference.
> Thus, LPDO does not increase the training cost compared to other baselines; rather, it enhances both computational efficiency and predictive effectiveness.
>
>
> ---
>
> > W4: Line 119: "target item" is confusing.
>
> Thank you for your insightful feedback. The subsection around Line 119 is intended to introduce diffusion models and their general application in recommender systems. The specific application of diffusion to the UBTP task, which involves predicting a sequence of k items, is detailed in Section 4.1. We will revise the manuscript to clarify this distinction and avoid potential confusion.
>
> ---
>
> > W5&W6: Line 158 and Line 198 is mistakenly referred.
>
> Thank you for pointing out the incorrect references. We will carefully review and correct these references in the revised version.
>
> ---
>
> > Q2: Line 153 mentioned that the paper follows [18] to also include history H as part of X0, which the model is trying to generate.
>
> Thank you for this insightful question about our design choice! The concatenation of history H and target z is first proposed by DiffuSeq [4] in the NLP task, and then DCRec [5] introduce it in the sequential recommendation task for conditioning diffusion models. This can better capture detailed sequential patterns and contextual information from the user’s history [5]. Based on this, we introduce causal attention to further capture the preference evolution. As shown in Fig.2, the reverse/generation process starts from $X_T$, where $H_T$ and $z_T$ are noised. $H_t$ in each diffusion steps can provide extra historical user information during the generation until obtaining clean $H_0$ and $Z_0$.
>
> Empirically, we observe that this concatenation-based conditioning significantly improves the guidance signal during generation compared to approaches that do not explicitly model the historical context in this manner (e.g., DreamRec [6], PreferDiff [7]). As a result, our method achieves more accurate and user-preferred recommendations.
>
> ---
>
> > Q3: For each $t$, the history $H$ is already a part of $X_t$. Why do we have the extra $H$ in $p_\theta(X_{t-1}| X_t, H)$?
>
> Thank you for the insightful comment. We would like to clarify that, in addition to the $H$ component within $X_t$, there is an extra $H$ used as direct conditioning, as illustrated in Fig.2. This additional conditioning serves as the input to the cross-attention layer and plays a crucial role in guiding the generation process. This strategy was first introduced in Stable Diffusion [8] and later adopted in DCRec [5] for recommendation tasks. We thank the reviewer for this insightful question and will enhance the explanation accordingly in the revised manuscript.
>
> ---
>
> > Q4: It seems that $L_{ListPref}$ is defined for a particular future time step j? How would the system combine all the k future time steps in the loss?
>
> Thank you for highlighting the potential ambiguity in our presentation of the listwise loss.
> Equation (11) was framed as “per‐step” $L_{\rm ListPref}(j)$ simply to keep the expression compact and to focus the reader’s attention on the mechanics of the listwise term at a single horizon $j$. In fact, the full listwise objective used in all our experiments is the sum of these per‐step losses over $j=1,\ldots,k$, exactly as shown (and implemented) in the model code and in Equation (16). By summing across all $k$ positions and employing our cross‑step attention blocks inside the Cross‑Conditional Diffusion Transformer, the model naturally receives joint gradient signals that enforce causal consistency between early and later predictions. In this way, the loss truly operates on the entire future sequence, not just on each step in isolation.
>
> ---
>
> > Q5: Eq.16 contains both reconstruction loss and list preference loss. I thought the reconstruction loss already contains the list preference loss.
>
> Thank you for raising this point. We would like to clarify that Eq.(16) is derived from Eq.(5), with the primary modification being the introduction of the list preference loss, a novel ranking objective specifically designed for the UBTP task, replacing the original ranking loss in Eq.(5). Meanwhile, the reconstruction loss and regularization loss are preserved. We appreciate this constructive feedback and will revise the manuscript accordingly to enhance clarity.
>
> [1] BERT: Pre-training of Deep Bidirectional Transformers for Language Understanding
>
> [2] Exploring the limits of transfer learning with a unified text-to-text transformer
>
> [3] Language Models are Unsupervised Multitask Learners
>
> [4] Diffuseq: Sequence to sequence text generation with diffusion models
>
> [5] Dual conditional diffusion models for sequential recommendation
>
> [6] Generate What You Prefer: Reshaping Sequential Recommendation via Guided Diffusion
>
> [7] Preference Diffusion for Recommendation
>
> [8] High-Resolution Image Synthesis with Latent Diffusion Models

---

> > ### Comment · Reviewer_WfsB · 2025-08-04
> >
> > Thank you for the response. I intend to keep my rating.

---

### Note · Authors · 2025-08-13

Dear AC and reviewers,

We sincerely appreciate the time and effort that you, our AC and reviewers, have devoted to reviewing and discussing our submission. In the following, we summarize the main contributions of our paper and address key issues raised during the rebuttal and discussion period.

**Main Contributions:**

- **Introduction of a New Task: User Behavior Trajectory Prediction (UBTP).** Our paper introduces a novel task, UBTP, which aims to model and predict user long-term behaviors in recommender systems. Unlike traditional sequential recommendation tasks that focus solely on predicting the next item, UBTP challenges a model to generate a coherent, ordered sequence of future interactions based on a user’s past behavior. This task better reflects real-world user decision-making processes and provides a richer framework for understanding user evolving preferences.

- **Proposal of a Listwise Optimization Approach for UBTP.** To address the UBTP task, we propose a listwise preference modeling method that enables coherent and preference-aware generation of user behavior trajectories by incorporating a Plackett–Luce supervision signal into the variational lower bound in diffusion models.

**Key Issues and Responses:**

- **Key innovation of LPDO (Reviewer WfsB).** The key advantage of our proposed LPDO method lies in its direct optimization of the listwise preference for item sequences (Fig.1(d)), rather than treating items independently as in pointwise methods. This listwise approach better captures inter-item dependencies and preserves the overall ranking quality of the sequence, resulting in improved recommendation performance.

- **Comparison with Baselines (Reviewer AK4H and EHty).** LPDO consistently outperforms strong baselines adapted from sequential recommendation tasks like SASRec, DiffuRec, and DCRec across all metrics (Tab.1, up to +30.61%), demonstrating its ability to capture evolving user preferences and long-term dependencies in the UBTP task. While Top-1 Luce’s Choice models (e.g., SASRec-AR) primarily focus on local preference signals, LPDO optimizes global sequence rankings, thereby achieving significant performance gains across all metrics (Tab.1).

- **Open-source Code (Reviewer ULMe).**
We plan to release the data and code after the paper is published.

Once again, we sincerely thank you for your feedback and comments throughout this insightful review process.

Best regards,

Authors of submission-24648

---

### Decision · Program_Chairs · 2025-09-17

**Decision:**

Accept (poster)

**Comment:**

The paper introduces User Behavior Trajectory Prediction (UBTP), which forecasts coherent multi-step user interaction sequences rather than only the next item. To address this task, the authors propose Listwise Preference Diffusion Optimization (LPDO), which integrates Plackett–Luce listwise supervision into diffusion training through a variational bound. The paper also introduces a new metric, SeqMatch, designed for strict trajectory-level evaluation. Experiments on multiple benchmarks demonstrate consistent improvements over both strong sequential baselines (e.g., SASRec, DCRec) and diffusion-based baselines (e.g., DiffuRec).

The strengths are the clear formulation of UBTP as a distinct task (Reviewer AK4H initially questioned but accepted after rebuttal), the principled integration of listwise ranking into diffusion ELBO (Reviewer EHty), and strong empirical results across datasets (Reviewer ULMe, Reviewer WfsB). Ablations provided during rebuttal clarified the contributions of the loss and causal transformer, which addressed novelty concerns (Reviewer WfsB).

The weaknesses are clarity and notation issues (e.g., “UBTM” instead of “UBTP” -- Reviewer EHty, Reviewer AK4H), performance degradation on longer horizons (Reviewer EHty), sensitivity to hyperparameters (Reviewer EHty), and initially incomplete baseline comparisons (Reviewer AK4H, later partly addressed). Reproducibility was also initially a concern due to lack of code release, though the authors committed to providing it (Reviewer ULMe).

During discussion, most concerns were clarified: novelty and ablations satisfied Reviewer WfsB; code/statistical issues were resolved for Reviewer ULMe; Reviewer AK4H raised but then accepted the motivation and baseline coverage; Reviewer EHty did not re-engage after rebuttal. Overall, the contribution is original, technically solid, and empirically convincing despite remaining presentation issues.

Final Recommendation: Accept